# Supercritical density fluctuations and structural heterogeneity in supercooled water-glycerol microdroplets

Sharon Berkowicz[1,7], Iason Andronis [1,7], Anita Girelli [1], Mariia Filianina [1], Maddalena Bin [1], Kyeongmin Nam [2], Myeongsik Shin [2], Markus Kowalewski [1], Tetsuo Katayama [3,4], Nicolas Giovambattista [5,6], Kyung Hwan Kim [2] & Fivos Perakis [1] ✉

Recent experiments and theoretical studies strongly indicate that water exhibits a liquid-liquid phase transition (LLPT) in the supercooled domain. An open question is how the LLPT of water can affect the properties of aqueous solutions. Here, we study the structural and thermodynamic properties of supercooled glycerol-water microdroplets at dilute conditions ($\chi_g$ = 3.2% glycerol mole fraction). The combination of rapid evaporative cooling with femtosecond X-ray scattering allows us to outrun crystallization and gain access to the deeply supercooled regime down to $T$ = 229.3 K. We find that the density fluctuations of the glycerol-water solution or, equivalently, its isothermal compressibility, $\kappa_T$, increases upon cooling. This is confirmed by molecular dynamics simulations, which indicate that the presence of glycerol shifts the temperature of maximum $\kappa_T$ from $T$ = 230 K in pure water down to $T$ = 223 K in the solution. Our findings elucidate the interplay between the complex behavior of water, including its LLPT, and the properties of aqueous solutions at low temperatures, which can have practical consequences in cryogenic biological applications and cryopreservation techniques.

The thermodynamic behavior of water is complex[1–4]. For example, in the liquid state and at $P$ = 1 bar, water exhibits anomalous maxima in density $\rho(T)$ (at $T$ = 277 K)[5], isobaric heat capacity $C_P(T)$ (at $T$ = 229 K)[6], and isothermal compressibility $\kappa_T(T)$ (at $T$ = 230 K)[7]. A natural explanation of the anomalous behavior of liquid and glassy water is given by the liquid-liquid phase transition (LLPT) hypothesis[8] which has received overwhelming support in recent years from experiments[6,7,9–12] and theoretical investigations[13–15]. In the LLPT scenario, liquid water exists in two liquid states at very low temperatures (at approximately $T$ < 220 K), low-density liquid (LDL) at low pressures, and high-density liquid (LDL) at elevated pressures.

In the $P$–$T$ plane, LDL and HDL are separated by a (liquid–liquid) first-order phase transition line at very low temperatures that ends at a liquid–liquid critical point (LLCP) located at approximately $P_c$ = 50–100 MPa and $T_c$ = 200–220 K[16–18]. According to this hypothesis, liquid water at $T > T_c$ and ambient pressure is a supercritical mixture of HDL/LDL fluctuating domains[3,8,18,19]. In particular, the presence of the LLCP in the $P$–$T$ phase diagram of water implies that the liquid must exhibit lines of maxima (in the $P$–$T$ plane) in $\kappa_T(T)$ and $C_P(T)$ at constant pressure ($P < P_c$). These lines of maxima in $\kappa_T(T)$ and $C_P(T)$ converge onto a single line in the $P$–$T$ plane as $P \rightarrow P_c$, defining the so-called Widom line—specifically, the Widom line

[1]Department of Physics, AlbaNova University Center, Stockholm University, SE-10691 Stockholm, Sweden. [2]Department of Chemistry, Pohang University of Science and Technology (POSTECH), Pohang 37673, Republic of Korea. [3]Japan Synchrotron Radiation Research Institute, Kouto 1-1-1, Sayo Hyogo 679-5198, Japan. [4]RIKEN SPring-8 Center, Kouto 1-1-1, Sayo Hyogo 679-5148, Japan. [5]Department of Physics, Brooklyn College of the City University of New York, Brooklyn NY 11210, USA. [6]The Graduate Center of the City University of New York, New York NY 10016, USA. [7]These authors contributed equally: Sharon Berkowicz, Iason Andronis. ✉e-mail: f.perakis@fysik.su.se

corresponds to the states in the $P-T$ plane at which the correlation length reaches a maximum value[20]. The location of the Widom line at ambient pressure has been observed experimentally, indicating that liquid water exhibits maxima in the correlation length at $T \approx 230$ K[7]. In particular, a maximum in $\kappa_T(T)$ and $C_P(T)$ have also been identified at $T \approx 230$ K at ambient pressure[6,7], implying that the LLCP in water must be located at slightly elevated pressures[18]. While the hypothesized LLCP in liquid water has been found in numerous molecular dynamics (MD) simulations using different realistic water models[15,21–25], it has so far eluded direct experimental verification.

The LDL-HDL fluctuations in liquid water at ambient pressure and low temperatures may have significant implications for the thermodynamic, structural, and dynamic properties of aqueous solutions. For example, at dilute concentrations, they should induce maxima in thermodynamic response functions, such as $C_P(T)$ and $\kappa_T(T)$, which are associated with fluctuations in enthalpy and density, respectively. LDL-HDL fluctuations may also affect the solvation of biomolecules both at ambient and supercooled conditions. The behavior of supercooled liquid and glassy aqueous solutions has wide importance in scientific and engineering applications, such as in cryopreservation techniques and the study of biological matter at low and cryogenic temperatures. Cryoprotectants are often utilized to minimize freeze damage, by interrupting the hydrogen-bond network of water and thereby preventing ice nuclei from forming and growing[26,27]. Interestingly, it has been suggested that the addition of solutes to water can affect the hydrogen-bond structure in a manner that resembles increasing pressure, i.e. by suppressing the tetrahedral hydrogen-bonded water structures associated with LDL[28–30].

The following natural questions arise from our discussion above: do the LLPT and/or the HDL/LDL fluctuations observed in pure water also exist in aqueous mixtures? If so, at what concentrations do the LLPT and LDL/HDL fluctuations develop in the solution? What role do they play in the behavior/properties of the solution? We note that MD simulations indicate that an LLCP may exist in ionic aqueous solutions, although the location of the critical point in the phase diagram can be shifted relative to pure bulk water due to the presence of the solutes and how these interact with water[30–34]. This observation is also consistent with experiments of hyperquenched aqueous LiCl solutions, where a polyamorphic transition from high-density to low-density glass was observed[35–38]. MD simulations and experiments in glassy water also suggest that an LLPT may occur in dilute aqueous solutions containing alcohols and/or biomolecules[39].

Glycerol-water mixtures of different concentrations have become prototypical systems to study whether the LLPT and/or HDL/LDL fluctuations can manifest in organic aqueous solutions[28,29,40–46]. Experiments show that, upon isobaric cooling, solutions with intermediate glycerol concentrations ($\chi_g$ = 13–19% glycerol mole fraction) develop noticeable changes in the liquid structure. These changes were originally believed to stem from a low-temperature LLPT in the solution, related to the HDL/LDL fluctuations in pure water[28,29,45]. However, further experiments and simulations attributed the low-temperature solution behavior at intermediate glycerol concentrations primarily to the formation of ice crystallites and freeze-concentration of the remaining solution[41–45,47]. Nonetheless, at least for very dilute glycerol-water solutions, it is expected that the LLCP location in the $P-T$ plane, ($P_c$, $T_c$), shifts continuously with the addition of glycerol. Indeed, MD simulations and experiments in glassy glycerol-water solutions suggest that the LLCP shifts towards lower temperatures and/or higher pressures in the presence of glycerol, e.g., ($T_c$ = 150 K, $P_c$ = 30–50 MPa) for $\chi_g$ = 12–15% glycerol mole fraction[40,42,43]. We note that experiments confirming the existence of an LLCP in supercooled liquid glycerol-water solutions have so far been missing due to the fast ice nucleation in the samples which is difficult to avoid experimentally[41–45,47,48].

In this work, we probe experimentally the liquid structure and thermodynamic properties of deeply supercooled microdroplets of glycerol-water solutions in the dilute regime ($\chi_g$ = 3.2% glycerol mole fraction) down to $T$ = 229.3 K, which is not accessible by conventional methods. Our goal is to address the following questions: how does the LLPT of water affect the properties of dilute glycerol-water mixtures? How does the presence of glycerol affect the local water network structure and the LDL-HDL fluctuations? In particular, how does the addition of glycerol shift the reported maximum in the compressibility of pure water? Our approach utilizes the rapid evaporative cooling technique[7,9] combined with femtosecond X-ray scattering at SACLA X-ray free-electron laser (XFEL) (see Fig. 1). The use of a large-area detector enables us to measure simultaneously the wide-and small-angle x-ray scattering domains (WAXS and SAXS) of the structure factor $S(q)$. In particular, from the SAXS measurements, we extract the isothermal compressibility and correlation length of the associated density fluctuations. In addition, we complement the experimental results with molecular dynamics (MD) simulations. The MD simulations allow us to study the properties of the solution, including $S(q)$, $\kappa_T$, and the local molecular structure, down to $T$ = 190 K ($\chi_g$ = 3.2% glycerol mole fraction).

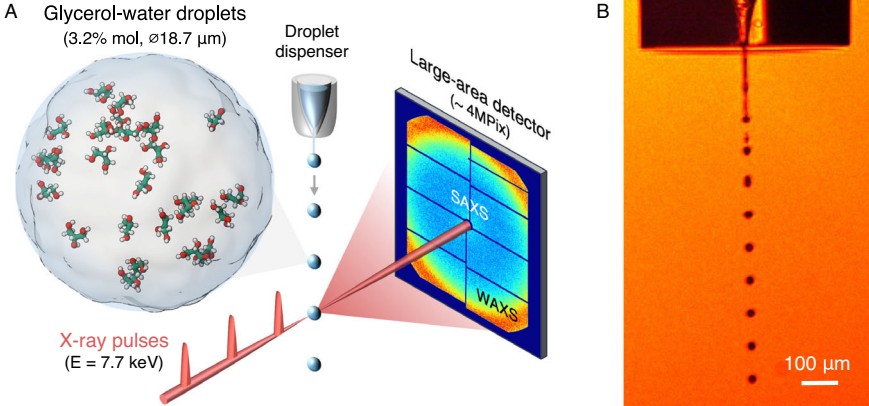

**Fig. 1 | The experimental setup combining rapid evaporative cooling of glycerol-water microdroplets with femtosecond X-ray scattering. A** A schematic overview of the experiment where glycerol-water solution ($\chi_g$ = 3.2% glycerol mole fraction) is supercooled by rapid evaporation in vacuum and probed by femtosecond X-ray pulses at SACLA X-ray free-electron laser (XFEL). The small- and wide-angle X-ray scattering (SAXS, WAXS) are measured simultaneously by using a large-area detector. **B** Microscope image of the glycerol-water droplets (18.7 $\mu$m in diameter) shown close to the liquid jet nozzle, recorded by stroboscopic illumination.

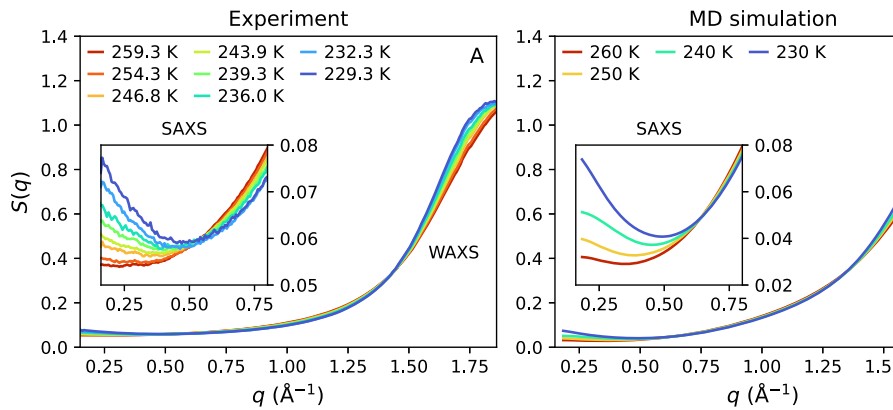

**Fig. 2 | The small- and wide-angle X-ray scattering (SAXS, WAXS) structure factor $S(q)$ of glycerol-water solutions ($\chi_g$ = 3.2% glycerol mole fraction) at different temperatures.** Comparison of $S(q)$ obtained by (**A**) femtosecond X-ray scattering experiments on microdroplets and (**B**) molecular dynamics (MD) simulations. The insets show magnifications of the SAXS region (note the different scale on the y-axis of the insets in (**A**) and (**B**)).

## Results

### X-ray structure factor

Figure 2A shows the experimental structure factor of supercooled droplets (18.7 $\mu$m in diameter) containing dilute glycerol-water solutions ($\chi_g$ = 3.2% glycerol mole fraction) at different temperatures probed by femtosecond X-ray scattering at SACLA XFEL. A large-area detector is used that provides a broad momentum transfer $q$-range spanning from SAXS ($q_{min} \approx 0.15$ Å$^{-1}$) to the first diffraction peak in WAXS ($q_{max} \approx 1.89$ Å$^{-1}$). We also measure the structure factor of bulk glycerol-water solutions ($\chi_g$ = 3.2% glycerol mole fraction) by table-top X-ray diffraction (XRD) at temperatures $T$ = 295−250 K. This allows us to measure the $S(q)$ over a larger $q$-range, up to $q \approx 6$ Å$^{-1}$; see Supplementary Information for details on the calculations of $S(q)$, reproducibility, and control of experimental conditions. Upon cooling, the first peak of $S(q)$ increases in magnitude and, in particular, it shifts continuously towards lower values of $q$. This observation is consistent with the behavior of the $S(q)$ in pure water, reflecting the increasing tetrahedral coordination of the water molecules upon cooling[7]. The SAXS region is emphasized in the inset of Fig. 2A, where a strong enhancement of the structure factor is observed with decreasing temperature. This SAXS enhancement is also consistent with previous experiments in supercooled pure water, which is attributed to the increase of density fluctuations and hence, to the increase of isothermal compressibility, upon cooling[7].

Interestingly, there is an isosbestic point in the $S(q)$ shown in Fig. 2A located at $q \approx 0.5$ Å$^{-1}$. While it is not evident whether there is an underlying physical/chemical reason for the existence of this isosbestic point in the $S(q)$, such an isosbestic point defines a T-independent wavevector q that can be used as a useful for future SAXS studies. The isosbestic point in $S(q)$ is a consequence of the increase in $S(0)$ upon cooling, which is due to the increase in thermal compressibility as the temperature of the solution is lowered. Since the value of $S(q_1)$ is practically T-independent, the increase of $S(0)$ upon cooling leads to an isosbestic point that barely shifts with decreasing temperature. An isosbestic point is also found in the $S(q)$ of pure water, located at $q \approx 0.4$ Å$^{-1}$[7]. MD simulations using TIP4P/2005 indicate an isosbestic point at $q \approx 0.25$ Å$^{-1}$ at $P$ = 1 bar, which shifts to $q \approx 0.4$ Å$^{-1}$ upon increasing pressure at $P$ = 1 kbar[49]. Accordingly, adding glycerol shifts the isosbestic point of the $S(q)$ slightly, towards higher $q$-values, at $q \approx 0.5$ Å$^{-1}$, which is consistent with the overall shift in the $S(q)$ of water, towards lower values of $q$, with the addition of glycerol. A similar shift of the isosbestic point of the $S(q)$ has been measured in the SAXS of NaCl-water solutions[50,51], resembling the trend found in computer simulations of pure water with increasing pressure[22].

Figure 2B shows the $S(q)$ of the glycerol-water solutions ($\chi_g$ = 3.2% glycerol mole fraction) calculated from MD simulations at temperatures $T$ = 230−260 K (see the Methods for details on the computer simulations). Qualitatively, the experimental and simulated structure factors show remarkable resemblance. In agreement with the experiments (Fig. 2A), the $S(q)$ obtained from the MD simulations exhibits a shift in the diffraction peak towards lower $q$-values and a pronounced enhancement of the SAXS intensity upon supercooling. On comparing the SAXS structure factors from experiments and MD simulations (see insets, Fig. 2), we note two main differences. Firstly, there is a slight shift in the $q$-position of the experimental and simulated isosbestic points of $S(q)$ (in the SAXS range). This is in agreement with previous observations in the $S(q)$ of pure water obtained from experiments/MD simulations[22]. Secondly, there is a small difference in the absolute intensity of the SAXS curves at small $q$. For example the minimum of $S(q)$ in Fig. 2A increases from $S(q) \approx 0.055$ to $S(q) \approx 0.060$ as the temperature decreases from $T \approx 260$ K to $T \approx 230$ K. Instead, in Fig. 2B, the minimum of $S(q)$ increases from $S(q) \approx 0.030$ to $S(q) \approx 0.040$ (for the same T-interval). This discrepancy is likely due to the difference in the $S(0)$ between experiments and MD simulations observed also for pure water[49], related to the fact that TIP4P/2005 underestimates the compressibility, (see Supplementary Note 5 for a direct comparison and detailed discussion). A small difference in the vertical offset and scaling of the SAXS curves can also result from the dilute-limit approximation used for the calculation of the $S(q)$ from the MD simulations (see Supplementary Note 1) or from experimental uncertainties arising from the background subtraction (see Supplementary Note 1.1).

Taking into account the difference in melting temperature of real water and TIP4P/2005 ($T_m \approx 250$ K), and thus comparing supercooled degrees instead of the absolute temperature is not sufficient by itself to account for the observed discrepancy (see Supplementary Supplementary Fig. 10). It has been shown that comparing the experimental data of pure water with MD simulations at elevated pressures yields a more accurate comparison of the SAXS regime and the corresponding compressibility[49]. This effect can be attributed to the fact that the location of the LLCP of the TIP4P/2005 model (see refs. 15,22–25,52) is shifted in pressure–temperature with respect to the LLCP estimation in real water[18].

### Wide-angle X-ray scattering (WAXS)

The structure factor first-peak position of the glycerol-water solution, $q_1(T)$, is shown in Fig. 3A for $T$ = 229.3-295 K. Included in Fig. 3A are the values of $q_1(T)$ extracted from (i) the WAXS measurements of microdroplets at SACLA XFEL ($T$ = 229.3−250 K; open black circles)

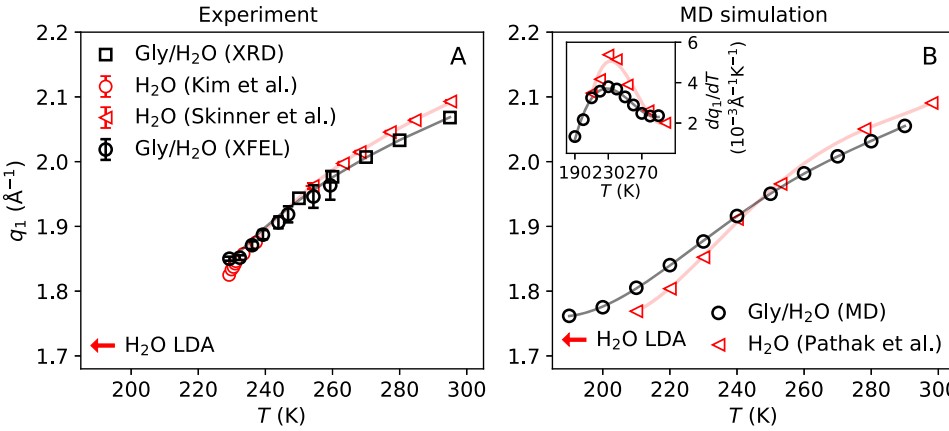

**Fig. 3 | Temperature-dependence of the structure factor first-peak position, $q_{1(T)}$, obtained from wide-angle X-ray scattering (WAXS) and MD simulations. A** Experimental data for the glycerol-water solutions ($\chi_g$ = 3.2% glycerol mole fraction) obtained from WAXS measurements using table-top X-ray diffraction (XRD) at moderately supercooled temperatures ($T \geq 250$–295 K, open black squares), and by femtosecond X-ray scattering of evaporatively cooled microdroplets ($T$ = 229.3–250 K, open black circles). For comparison, we include the experimental $q_1(T)$ of pure liquid water (open red circles and triangles) from refs. 7,53. **B** $q_1(T)$ obtained from MD simulations of glycerol-water solutions ($\chi_g$ = 3.2% glycerol mole fraction, open black circles) and pure liquid water from ref. 49 (open red triangles). Solid lines are smoothing spline fits. The inset in (**B**) shows the temperature-derivative $dq_1/dT$ for the glycerol-water solution (open black circles) and pure water (open red triangles). The maximum in $dq_1/dT$ occurs at $T$ = 228 K for the glycerol-water solution and at $T$ = 233 K for pure water, respectively. In both figures, error bars smaller than the size of the corresponding data points and the red arrows are used to indicate the $q_1$ of low-density amorphous (LDA) ice, according to experiments[84] at $T$ = 80 K, and simulations[85] at $T$ = 80 K. Error bars indicate the standard error obtained from the gaussian fit to the first $S(q)$ peak.

and (ii) the bulk solution using table-top XRD ($T$ = 250-295 K; open black squares). With conventional cooling methods, i.e., with cooling rates of ~1 K/s, the lowest temperature accessible to the glycerol-water solution before crystallization intervenes is $T \approx 250$ K (3.2% glycerol mole fraction). The use of evaporative rapid cooling of microdroplets allows us to extend the experimental data down to $T$ = 229.3 K (see Supplementary Note 3 for details on the temperature estimation). Within the measured temperature range, $T$ = 229.3–295 K, $q_1(T)$ decreases continuously upon cooling. In addition, the experimental data indicates that the corresponding rate of change, $dq_1(T)/dT$, increases upon cooling. This trend is consistent with the temperature dependence of $q_1(T)$ measured in pure water (open red triangles and circles from refs. 7,53). In the case of pure water, it has been shown that the decrease of $q_1(T)$ upon cooling is correlated with an increase in the tetrahedral order of the water molecules[7]. Interestingly, Fig. 3A shows that at high temperatures, approximately $T > 250$ K, the addition of glycerol reduces the values of $q_1$ at a given temperature, while at $230 < T < 250$ K, the $q_1(T)$ of pure water and the glycerol-water solution nearly overlap. This suggests that, at least for $T > 250$ K, glycerol promotes a comparatively more open tetrahedral arrangement of solvent molecules, relative to bulk water. The ability of glycerol to form multiple hydrogen bonds can play a key role, enabling glycerol to incorporate itself within the hydrogen-bond network of water.

Figure 3B shows the $q_1(T)$ for the glycerol-water solution calculated from our MD simulations (open black circles) as well as the corresponding results obtained for bulk TIP4P/2005 water reported in ref. 49. Overall, the results from the MD simulations shown in Fig. 3B are in semi-quantitative agreement with the experimental data (Fig. 3A); see also Supplementary Fig. 6A. While the experimental data is limited to $T \geq 229.3$ K due to rapid crystallization, the lack of crystallization in the MD simulations allows us to explore the behavior of $q_1(T)$ at lower temperatures. As shown in Fig. 3B, two temperature regimes can be identified: a high-temperature regime ($T > 240$ K) where the $q_1(T)$ of the glycerol-water solution is lower than the $q_1(T)$ of pure water, and a low-temperature regime ($T < 240$ K) where the $q_1(T)$ of the glycerol-water is higher than that of pure water. It follows that at low temperatures, the addition of glycerol shifts the first peak of $S(q)$ towards larger values, relative to the pure water case, suggesting that

at $T < 240$ K glycerol may partially suppress the local tetrahedral structure of water.

We conclude this section by noticing that the MD simulation of glycerol-water solutions show an inflection point in $q_1(T)$ at around $T$ = 231 K, yielding a maximum in the temperature-derivative $dq_1/dT$ (see the inset of Fig. 3B). This temperature is practically identical to the Widom line temperature in pure water at ambient pressure reported in ref. 21. Importantly, in the case of pure water, a maximum in the experimental $dq_1/dT$ has also been reported at $T \approx 230$ K which coincides with the experimental Widom line temperature of pure water at ambient pressure[7] (based on the maxima in correlation length).

## Small-angle X-ray scattering (SAXS)

Next, we focus on the features of $S(q)$ at $q < 0.5$ A$^{-1}$. An enhancement of the $S(q)$ in the SAXS region upon supercooling (Fig. 2) is associated with the presence of density fluctuations[7,50,51]. Indeed, the isothermal compressibility $\kappa_T$, which is the thermodynamic response function that quantifies the density fluctuations in the system, is given by[54]

$$\kappa_T = \frac{S(0)}{n k_B T}. \tag{1}$$

where $n$ is the average molecular number density in the solution estimated from the corresponding total density and average molecular mass (see Supplementary Note 2); $k_B$ is the Boltzmann constant. The correlation length associated with the density fluctuations, $\xi(T)$, can also be extracted from the SAXS measurements (see Methods and Supplementary Fig. 5).

Figure 4A shows the experimental $\kappa_T(T)$ of glycerol-water solution obtained using Eq. (1) (black open circles). Also included is the $\kappa_T(T)$ for bulk water reported in the experiments of ref. 7 (empty red triangles). In both cases, $\kappa_T(T)$ increases rapidly upon cooling. A similar trend is seen in the correlation length $\xi(T)$ (see Supplementary Fig. 4B). Compared to pure water (solid red line, Fig. 4A)[55], the isothermal compressibility for the glycerol-water solution is significantly lower over the entire experimental temperature range $T$ = 229.3–295 K. This indicates that even though the glycerol-water system exhibits anomalous density fluctuations that increase upon cooling, such fluctuations are less pronounced than in pure water. It follows that the

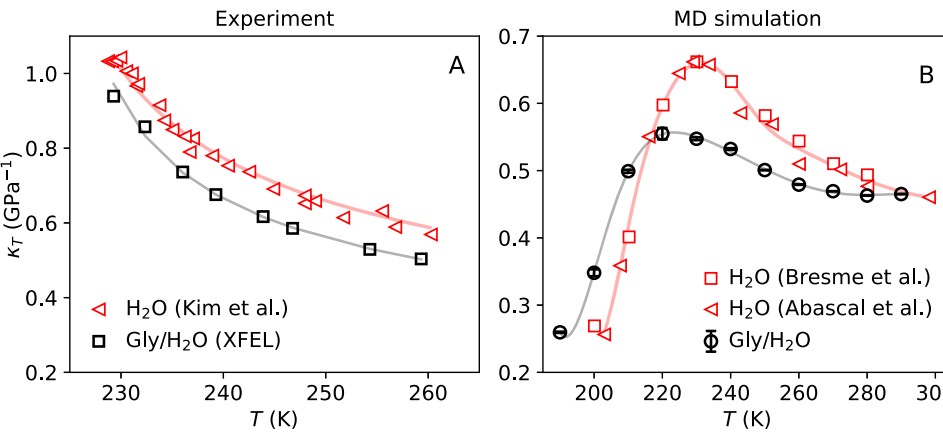

**Fig. 4 | Temperature-dependence of the isothermal compressibility $\kappa_T$.**
**A** Isothermal compressibility, $\kappa_T$, of glycerol-water solutions ($\chi_g$ = 3.2% glycerol mole fraction, black open circles) calculated from the structure factor obtained using the femtosecond X-ray scattering (SAXS). Also included is the $\kappa_T$ of pure water from the SAXS experiments reported in ref. 7 (open red triangles). Solid lines are power-law fits to the data using Eq. (2)[55] (see also Table 1). **B** The $\kappa_T(T)$ of the glycerol-water solution ($\chi_g$ = 3.2% glycerol mole fraction, black open circles) and pure TIP4P/2005 water obtained from MD simulations (open red triangles[21] and squares[86]). A maximum in $\kappa_T(T)$ is found in the MD simulations of both pure TIP4P/2005 water ($T$ = 234 K) and glycerol-water solutions ($T$ = 223 ± 1 K). The solid lines are spline interpolations to the data. Note the difference in the scale of the x and y axis between the panels. Error bars indicate the standard error.

presence of glycerol partially disrupts the collective density fluctuations of water by (i) direct interactions with the water molecules, and/or (ii) by effectively inducing confinement effects which may suppress the size of the high/low-density transient domains.

We fit the experimental values of $\kappa_T(T)$ and $\xi(T)$ to power-law relations[54,55], specifically,

$$\kappa_T(T) = \kappa_{T,0}\, \epsilon^{-\gamma} \quad \text{and} \quad \xi(T) = \xi_0\, \epsilon^{-\nu}, \qquad (2)$$

where $\epsilon = (T - T_s)/T_s$ and $T_s$ is the temperature at which each property diverges. The $\xi_0$ and $\kappa_{T,0}$ are constants; $\nu$ and $\gamma$ are the associated power law exponents. The fit to the experimental values of $\kappa_T(T)$, using Eq. (2), is included in Fig. 4A (black solid line). The corresponding exponent and characteristic temperature are $\gamma$ = 0.36 ± 0.02 and $T_{s,\kappa}$ = 224 ± 1 K. Similar results hold for the correlation length (see Supplementary Fig. 4B), with corresponding exponent and temperatures being $\nu$ = 0.26 ± 0.1 and $T_{s,\xi}$ = 221 ± 7 K. The power-law fitting parameters are given in Table 1 and are close to the corresponding values reported from experiments in pure water[55]. If the power law behavior in $\kappa_T(T)$ and $\xi(T)$ was due to an underlying (liquid–liquid) critical point, the power-law exponents, $\nu$ and $\gamma$, should increase and reach maximum values as the system approaches the critical pressure[54,56,57]. In the case of the Ising model, the maximum values for the corresponding exponents are $\nu$ = 0.63 and $\gamma$ = 1.24 with a ratio of $\nu/\gamma$ = 0.5[54,56,57]. Previous analysis of the power law exponents of those obtained experimentally supercooled water indicate that ratio of the exponents, $\nu/\gamma$ = 0.65, which is relatively close to the ratio $\nu/\gamma$ = 0.51 that would be expected exactly at a critical point. Analysis of the $\gamma$ exponents obtained from MD simulations indicate that the various models examined (TIP4P/2005, SPCE, E3B3 and iAMOEBA) underestimate the $\gamma$ values compared to the experiment[55].

**Table 1 | Power-law fitting parameters for the $\kappa_T(T)$ and $\xi(T)$ of the glycerol-water solution ($\chi_g$ = 3.2% glycerol mole fraction) based on the experimental data shown in Fig. 4 and Supplementary Fig. 4B**

| Sample | $\gamma$ | $\nu$ | $\nu/\gamma$ |
|---|---|---|---|
| Glycerol-water ($\chi_g$ = 3.2 mol%) | 0.36 ± 0.02 | 0.26 ± 0.1 | 0.73 ± 0.4 |
| Pure water[55] | 0.40 ± 0.01 | 0.26 ± 0.3 | 0.65 |
| Ising model[54] | 1.2 | 0.6 | 0.5 |

The fitting parameters are defined in Eq. (2).

Here we observe that the $\gamma$ exponent in glycerol-water is lower than expected for the LLCP, indicating that the system is in the supercritical region and that this apparent critical behavior is associated with approaching the Widom line upon cooling[55]. In addition, the $\gamma$ value for the glycerol-water solution ($\gamma$ = 0.36 ± 0.02) is lower than pure water ($\gamma$ = 0.40 ± 0.01) indicating possibly that glycerol partially suppresses the critical fluctuations, which is consistent with the relative reduction in the compressibility.

An important observation from Fig. 4A is the lack of any clear maximum in the $\kappa_T(T)$ of the glycerol-water solution (3.2% glycerol mole fraction) over the temperature range studied, $T$ = 229.3–295 K (black solid line). Similarly, the $\xi(T)$ data of the glycerol-water solution, does not indicate a maximum (magenta solid line in Supplementary Fig. 5B). This is in contrast to the case of pure water, where experiments show a maximum in $\kappa_T(T)$, $\xi(T)$, and specific heat capacity $C_P(T)$, all at $T \approx 230$ K (the maxima in $\kappa_T(T)$ is mild but can be observed in Fig. 4, red empty triangles; reproduced from ref. 7). In the case of pure water, the maxima in these properties are associated with the system crossing the Widom line upon isobaric cooling at ambient pressure[6]. We note, however, that the number of data points for the glycerol-water solution in Fig. 4A is smaller than the number of points available for pure water[7]. Therefore, more experiments are needed, at approximately $T$ = 230 K, to confirm the absence of an $\kappa_T$-maximum in the glycerol-water solution ($\chi_g$ = 3.2% glycerol mole fraction). Nevertheless, it is clear that adding glycerol shifts the values of $\kappa_T(T)$ towards lower temperatures, and, as discussed below, the results from MD simulations strongly suggest that such a maximum is pushed to $T < 230$ K.

Figure 4B shows the $\kappa_T(T)$ of the glycerol-water solution and pure water calculated from MD simulations which allows us to get insight into lower temperatures. As discussed in the Methods section, $\kappa_T(T)$ is obtained by calculating the volume fluctuations in the system. The MD simulations reveal a maximum in the $\kappa_T(T)$ of the glycerol-water solution at $T$ = 223 ± 1 K; see Fig. 4B (black open circles). A $\kappa_T$-maximum is also found in bulk water (red empty triangles; see below). We note that the overall magnitude of the isothermal compressibility enhancement upon cooling is suppressed in the MD simulations in comparison to the experiment (see also Supplementary Fig. 6B), a known limitation of the TIP4P/2005 and most water models[58].

In order to further explore whether the experimental $\kappa_T(T)$ data provide any indications of a maximum in the $\kappa_T(T)$, we analyze the goodness of fit for the power-law model, based on the coefficient of

determination $R^2$. The power-law model predicts a divergence at $T = T_s$, where the $\kappa_T(T)$ would be infinite. Approaching the Widom line, it is expected that the $\kappa_T(T)$ would deviate from the power-law prediction in the proximity of the $\kappa_T(T)$ maximum[55]. Supplementary Fig. 8A shows the power-law fit for the experimental $\kappa_T(T)$ data, including the whole temperature range (dashed line) and by excluding the $\kappa_T$ data point at $T = 230$ K (solid line). Based on the $R^2$, we observe that the best fit to power law is obtained by excluding the $\kappa_T$ data point at $T = 230$ K (solid line). Hence, the data point for $\kappa_T(T)$ at $T = 230$ K deviates from the power-law behavior. We validate this approach by performing a similar analysis on the MD simulations, shown in Supplementary Fig. 8B. Again, we observe a similar behavior, whereby excluding the $\kappa_T(T)$ data at $T \leq 230$ K (solid line) provides a significantly better agreement ($R^2 = 0.992$) with a power law. The fit to all values of $\kappa_T(T)$ including the $T = 230$ K data point is shown by the dashed line ($R^2 = 0.936$). This result indicates that the $\kappa_T(T)$ data deviate from the power law at $T = 230$ K. We note that the data at $T = 220$ K deviate even further from the power-law behavior, as this is the temperature where the $\kappa_T(T)$ maximum is observed.

It should be noted here, that the deviation at $T \leq 230$ K appears larger for the MD simulation than in the experiment, likely due to limitations of the MD model. TIP4P/2005 model underestimates the amplitude of the $\kappa_T(T)$ maximum for pure water. Indeed, a shift in pressure results in better agreement between the results of MD simulations of TIP4P/2005 water and experiments which can be explained by considering that the location of the liquid–liquid critical point in TIP4P/2005 water is shifted, in the $P$–$T$ plane, relative to the corresponding location of the liquid–liquid critical point of real water. This effect is also seen in the broader $\kappa_T(T)$ maximum of TIP4P/2005 water, compared to experiments[21], which implies that any deviation observed for the power-law behavior would be more significant for the simulation than the experimental data, as observed in Supplementary Fig. 8B.

A comparison of the MD simulations results shown in Fig. 4B indicates that the addition of glycerol (i) reduces the value of the maximum in $\kappa_T(T)$ and (ii) shifts the $\kappa$-maximum to lower temperatures. In particular, point (ii) is fully consistent with the experimental data shown in Fig. 4A, where the $\kappa(T)$ of the glycerol-water is always lower than that of pure water. This observation implies that adding glycerol shifts the Widom line of water to lower temperatures at ambient pressure. This also suggests that adding glycerol has a similar effect on water as increasing the pressure, disrupting the hydrogen-bond network of water[28]. However, we note that in pure water, increasing pressure also increases the maximum value of $\kappa_T$ since the system is brought closer to the critical point pressure of water[55]. The observed smaller magnitude of the $\kappa_T$ for the glycerol-water solution, compared to pure water, can be rationalized by considering that, unlike the external force of pressure, the water structure is disrupted locally by the presence of the glycerol molecules *within* the solution. The intercalating glycerol molecules can cause an additional confinement effect which can limit the size of correlated water domains and thereby suppress the magnitude of the density fluctuations.

### Local structure index
To explore the local structure of water within the glycerol-water solution, we calculate the local structure index (*LSI*) of the water molecules in the system. The *LSI* describes the local structure around a central water oxygen atom in terms of the distances between neighboring oxygen atoms (see Methods section). High values of *LSI* indicate ordered structures with well-defined first and second hydration shells, while low values of *LSI* indicate disordered, collapsed structures with molecules populating the interstitial shell, in between the first and second hydration shells[59–61]. MD simulations of pure liquid water at room temperature and $P = 1$ bar exhibit a bimodal *LSI* distribution[14,61,62]: highly tetrahedral LDL-like molecules result to large values of *LSI* while

HDL-like molecules, characterized by populated first-interstitial shells, exhibit low values of *LSI*. Figure 5A shows the distribution of *LSI* values calculated from the MD simulations of the glycerol-water solution at different temperatures. To obtain well-defined *LSI* distributions, we remove the molecular intermolecular vibrational effects due to thermal fluctuations and evaluate the *LSI* at the inherent structures (potential energy minima) of the system (see Methods). We find that, as reported in MD simulations of pure water[14,61,62], the glycerol-water solutions exhibit temperature-dependent bimodal *LSI* distributions, with a local minimum at $LSI \approx 0.114$ Å$^2$. The peak located at $LSI < 0.114$ Å$^2$ corresponds to HDL-like water molecules and shrinks upon cooling, while the second peak located at $LSI > 0.114$ Å$^2$ corresponds to LDL-like water molecules and becomes more pronounced with decreasing temperatures. Figure 5B shows the fraction of HDL-like ($LSI \leq 0.114$ Å$^2$) and LDL-like molecules ($LSI > 0.114$ Å$^2$) calculated from Fig. 5A. At $T \approx 232$ K, the glycerol-water solution is composed of an equal amount of LDL- and HDL-like water molecules; at higher (lower) temperatures, the majority of the water molecules are HDL-like (LDL-like).

Snapshots of the glycerol-water solution extracted from our MD simulations are included in Fig. 5C for the cases $T = 190$ K (left), $T = 230$ K (center) and $T = 270$ K (right). Here, the glycerol molecules are colored in white, while the HDL- and LDL-like water molecules are shown in red and blue colors, color-coded based on their classification as HDL-like (with $LSI \leq 0.114$ Å$^2$) and LDL-like (with $LSI > 0.114$ Å$^2$), respectively. As expected, the system at $T = 190$ K is characterized by large cohesive LDL-like (blue) domains containing scattered glycerol molecules; a small number of residual HDL-like water molecules are also observed. The opposite scenario is observed at high temperatures; at $T = 270$ K, the system is composed mostly of HDL-like (red) molecules surrounded by scattered glycerol molecules and LDL-like domains. We note that, at $T = 270$ K, the relative difference between the HDL-like and LDL-like fraction of molecules is less pronounced than at $T = 190$ K, as observed before for pure water[61]. At $T = 232$ K, on the other hand, there are nearly equal fractions of HDL-like and LDL-like waters that form highly interpenetrating networks. The rather large percolation of the LDL and HDL domains throughout the system could reflect maximal fluctuations in the proximity to the Widom line. This observation closely coincides with previous simulations of pure water with the TIP4P/2005 model, where a 1:1 distribution between HDL- and LDL-like molecular species was observed at $T \approx 233$ K[61]. At $T = 230$ K, we observe that the LSI does not differ whether one examines the bulk water or that in the proximity of the glycerol (see Supplementary Supplementary Fig. 13). It should be noted here, as shown in ref. 63, that these results can depend on the local order parameter used to examine the nature of water in the hydration layer. Our study based on the LSI order parameter suggests that at $T = 230$ K, the local glycerol environment consists 1:1 of HDL/LDL water molecules, which reversely indicates that the system is isocompositional, i.e. LDL and HDL have the same concentration of glycerol, as suggested previously[28].

### Discussion
In this work, we present results from femtosecond SAXS and WAXS experiments, and complementary MD simulations of dilute glycerol-water solutions ($\chi_g = 3.2\%$ glycerol mole fraction) over a wide range of temperatures, $T = 229.3–295$ K. By using rapidly evaporating micro-droplets in vacuum (18.7 $\mu$m in diameter), we study the glycerol-water solutions deep in the supercooled liquid regime ($T = 229.3$ K), thereby avoiding crystallization. Overall, we find a good agreement between the SAXS/WAXS experiments and MD simulations. We observe that:

(i)  In the WAXS regime, the structure factor first-peak position $q_1(T)$ shifts towards lower values of momentum transfer $q$ as the temperature decreases [Figs. 2, 3]. A similar temperature effect on $q_1(T)$ has been reported for pure water[7,9]. This implies that, as in the case of pure water, the local arrangements of the water molecules within the glycerol-water solution become more

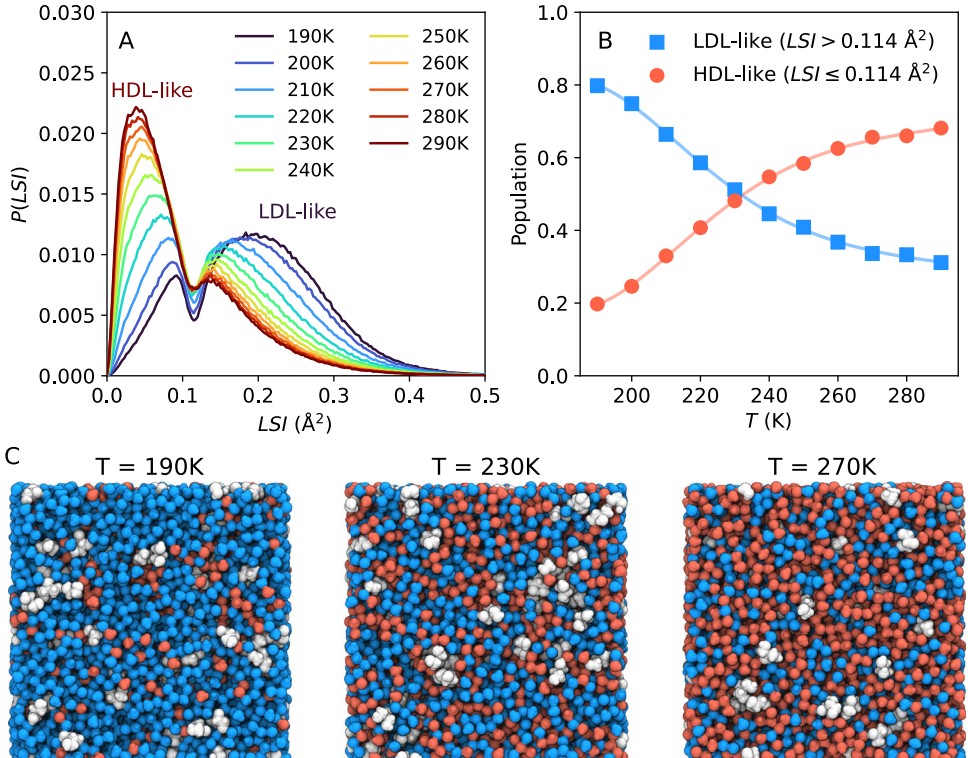

**Fig. 5 | The inherent local structure index (LSI) of water from MD simulations of glycerol-water solution at different temperatures. A** Probability distributions of the inherent *LSI* for the water molecules shown for different temperatures. The distribution is bimodal corresponding to HDL-like water molecules at *LSI* < 0.114 Å² and LDL-like water molecules for *LSI* > 0.114 Å². **B** The temperature dependence of the *LSI* populations, indicating a decrease of the HDL-like population (red circles, *LSI* ≤ 0.114 Å²) and an increase of the LDL-like population (blue squares, *LSI* > 0.114

Å²) upon cooling. The population crossing occurs at approximately *T* = 232K. Solid lines correspond to smoothing spline fits. **C** Snapshots of the MD simulation box at *T* = 190 K (left), *T* = 230 K (center) and *T* = 270 K (right). Here, the water molecules are colored according to their inherent *LSI* value with HDL-like water in red (*LSI*≤0.114 Å²) and LDL-like in blue (*LSI* > 0.114 Å²), while glycerol molecules are shown in white. The simulation boxes were rendered using VMD[87].

ordered (tetrahedral) upon cooling [Fig. 5]. Interestingly, the temperature-derivative $dq_1/dT$ obtained from the MD simulations of the glycerol-water solution and pure water exhibit a maximum at a similar temperature, $T \approx 230$ K. In the case of pure water, this temperature coincides approximately with the Widom line temperature (at ambient pressure)[7].

(ii) At very low $q$, in the SAXS regime, both experiments and MD simulations show a strong increase in the structure factor of the glycerol-water solution upon cooling. This implies that there is an anomalous increase in density fluctuations in the solution with decreasing temperature. From the SAXS experiments, we calculate the isothermal compressibility $\kappa_T(T)$ of the glycerol-water solution, and the correlation length of the associated density fluctuations, $\xi(T)$. Both quantities increase anomalously upon cooling following a power law. The corresponding power-law exponents are not large enough to justify the existence of a (liquid–liquid) critical point at ambient pressure. Instead, our results indicate that the microdroplets are in the supercritical state and are fully consistent with those from experiments performed for pure water[7,55].

An important experimental result of our study is that the addition of glycerol reduces the isothermal compressibility and the associated density fluctuations of pure water (at a given temperature); this could be, for example, due to the direct interactions (hydrogen-bonds formation) between the glycerol and water molecules, and/or because the glycerol molecules may limit the size of the density fluctuations (confinement effect). Additionally, our results indicate that adding glycerol shifts both the $\kappa_T(T)$ and $\xi(T)$ of pure water towards lower

temperatures. We note that the $\kappa_T(T)$ of pure water exhibits a maximum at $T \approx 230$ K while, instead, our studies on glycerol-water solutions ($\chi_g = 3.2\%$ glycerol mole fraction)) suggest that such a maximum, shifts to lower temperatures due to the presence of glycerol. Extending our experiments to lower temperatures, $T < 230$ K, to directly observe the maximum in $\kappa_T(T)$ is highly non-trivial. One possible avenue to extend the experimentally accessible temperatures to significantly lower values would be to utilize the high repetition rates of superconducting XFELs (like the European XFEL or LCLS-II), which allow the acquisition of data at kHz to MHz rates (vs 50Hz at SACLA). This approach would enable orders of magnitude more statistics even at very low hit-rates, which can be especially useful for accessing the regime $T < 230$ K, where the hit-rate is on the order of 0.1%. This comes with challenges related to operating the liquid jet at kHz to MHz rates, as well as big data volumes, which have been solved already for serial crystallography[64]. An alternative approach to explore the existence of an LLCP at elevated pressure would be to prepare high-density amorphous ice samples with different glycerol-water concentrations and perform a temperature-jump experiment (as in refs. 6,10,12). Possible challenges here relate to establishing accurately the trajectory across the pressure–temperature phase space and creating thin ice samples to ensure homogeneous heating.

The effects of glycerol on the $\kappa_T$-maximum of water are nicely demonstrated by our MD simulations which allows us to extend the temperature range accessible to the glycerol-water solution down to $T = 190$ K. The MD simulations reveal that a maximum in the $\kappa_T(T)$ of the glycerol-water solution indeed exists, and that it is shifted to lower temperatures relative to pure water. Specifically, the $\kappa_T$-maximum temperatures are $T = 223 \pm 1$ K for the glycerol-water solution and pure

water. In the case of pure water, the $\kappa_T$-maximum observed in experiments ($T \approx 230$ K[7]) and MD simulations (TIP4P/2005 $T \approx 234$ K[21]) was linked to density fluctuations between HDL/LDL domains, and the $\kappa_T$-maximum itself was identified as the Widom line[6,7,55,58] emanating from a LLCP. The same interpretation applies to the water domains within the glycerol-water solution; the role of glycerol is just to shift such LDL/HDL fluctuations to lower temperatures. Our results are further consistent with the previous MD simulations of ion-water mixtures, which indicate that the location of the LLCP in the phase diagram is shifted relative to pure bulk water due to the presence of the solutes[31–34].

The analysis of the *LSI* order parameter, based on the MD simulations, is also consistent with the presence of LDL and HDL domains in the glycerol-water solution. Specifically, we found that the fraction of LDL-like water molecules within the mixture increases upon cooling. Importantly, the MD simulations show that at $T \approx 232$ K the system is composed of equal amounts of HDL- and LDL-like molecules, similar to the the case of pure water. This observation indicates that even though the $\kappa_T$-maximum, reflecting collective density fluctuations, is shifted to lower temperatures in the glycerol-water solution ($T = 223 \pm 1$ K), the local water structure in the mixture follows the temperature dependence of pure water. This interpretation is consistent with the observed maximum of the temperature-derivative $dq_1/dT$ at $T \approx 231$ K, as it is also a local structural probe. That can be an indication that the line in the phase space of the glycerol-water solution defined by the $\kappa_T$-maxima deviates from those defined by structural observables, such as the $dq_1/dT$ and the HDL/LDL populations extracted with the *LSI*. The lines of maxima of different thermodynamic response functions, as well as dynamic and structural observables, can deviate from each other in the supercritical of the solution[65]. These lines of maxima of the different properties should, however, converge upon approaching the critical point into a single line, the Widom line, defined as the line of the maxima of the correlation length. Therefore, our data indicate that the system is in supercritical conditions at this region of the phase space (ambient pressure and approximately $T > 229$ K) and that if an LLCP exists in the glycerol-water mixture, it is expected to be located at higher concentrations and/or higher pressures than those studied here. This observation is in accordance with previous studies starting from glassy samples, which indicate an LLCP in the glycerol-water system at ($T$, $\chi_g$, $P$) = (150 K, 13.5%, 45 MPa).

In conclusion, our results shed light on the influence of glycerol on the local structure of supercooled water and provide evidence that the two-liquid framework of water can be extended to describe the thermodynamic properties of solutions. Alternatively, our findings show how the properties of pure water, and its underlying LLCP, may affect the properties of aqueous solutions. We find that, even at dilute conditions, the presence of glycerol can partially suppress the collective density fluctuations of pure water and shift its LLCP towards lower temperatures. The suppression of density fluctuations observed here can be linked to glycerol's cryoprotectant ability to frustrate the fluctuations associated with the formation of the critical nucleus preceding crystallization of the system[66]. The knowledge of the changes in the local water structure and nanoscale fluctuations, with increasing glycerol content, as provided in the present study, are then potentially crucial to design the appropriate cryoprotectant mixtures. Our results may therefore be important to advance cryopreservation techniques and the design of cryoprotectants that better prevent ice nucleation, a key challenge in cryopreservation. By tuning glycerol concentrations, or combining it with other cryoprotectants like DMSO or ethylene glycol, formulations can be optimized to maximize the suppression of crystallization while maintaining low toxicity[27]. This is particularly relevant for biological applications where ice formation can damage cells and can be beneficial for cryopreserving complex tissues or organs, where ice formation must be avoided[26]. Additionally, understanding the impact of

**Table 2 | XFEL experiment parameters**

| | |
|---|---|
| Photon energy | 7.7 keV |
| Pulse energy | ≈ 0.5 mJ |
| Pulse length | <10 fs |
| Repetition rate | 30 Hz |
| Beam focus size (FWHM) | $4.5 \times 3$ $\mu m^2$ (horizontal × vertical) |
| MPCCD octal SWD detector | $4 \times 2$ units, $1024 \times 512$ pixels per unit |
| Pixel size | 50 $\mu m$ |
| Accessible $q$-range | $q \approx 0.15$-$1.89$ Å$^{-1}$ |

glycerol on water local structure at elevated pressure would benefit the development of high-pressure cryopreservation techniques in combination with glycerol-based solutions that could further enhance control over freezing processes[67].

## Methods

### X-ray free-electron laser (XFEL) measurements

The XFEL experiments were performed at the BL3 beamline (EH2 hutch) at the SPring-8 Angstrom Compact free electron LAser (SACLA) in Japan (proposal no. 2022B8033). The experimental parameters are summarized in Table 2.

### Sample and droplet setup parameters

For the glycerol-water solution droplets setup, we follow a protocol similar to that employed in ref. 7 to study the properties of supercooled droplets of pure water. Here, we prepare liquid droplets of 18.7 $\mu m$ in diameter composed of glycerol-water solutions of $\chi_g = 3.2\%$ glycerol mole fraction (14.5 wt%). We used MilliQ water and glycerol from Sigma-Aldrich (prod. no. G9012). The droplets are supercooled by the rapid evaporative cooling technique described in refs. 6,7,9,68. The droplets were generated using a piezo-driven microdispenser device with an orifice diameter of 10 $\mu m$- (MJ-ATP-01-10-8MX from Microfab), and modulated at frequency 156–157 kHz. A back pressure of 2.5–2.8 bar nitrogen gas was applied to the sample solution container to pump the liquid through the dispenser, creating a train of equidistant droplets with 85 $\mu m$ center-to-center distance (see Fig. 1. The droplets were injected vertically downwards into a vacuum chamber at vacuum pressure 1.6 Pa and the jet was captured by a cryotrap. The sample vacuum chamber was directly connected to the X-ray flight tube downstream and with an 8-inch exit kapton window (125 mm thickness) at ≈ 95 mm from the X-ray interaction point. A beam stop (Tungsten-Aluminum-Graphite cylinder, 3 mm in diameter and 13 mm in length) was glued to the inside of the exit window to block the direct beam.

To study the properties of glycerol-water solution at different temperatures, data was collected from droplets at different distances from the dispenser tip. The position of the dispenser was controlled using a manipulator (VAb Vakuum-Anlagenbau GmbH). The temperature of the droplets, which decreases as they travel in vaccuum (evaporative coolingin the range $T = 229$–$259$ K) was estimated using the Knudsen evaporation theory and thermodynamic properties of glycerol-water mixtures. The Knudsen evaporation model for droplet temperature estimation has been examined in detail previously, both experimentally based on XFEL[7,9] and Raman[69] measurements, as well as from MD simulations[70] (see also Supplementary Note 3).

To monitor the position and characterize the droplet stream during the measurements, an optical microscope was focused on the X-ray focal point in the direction perpendicular to the optical axis. The droplet size and droplet-droplet distance were calibrated from the microscope images (see example in Fig. 1) close to the dispenser tip, and from the known outer diameter of the dispenser housing (565.4 $\mu m$), as determined from high-resolution optical microscopy.

## WAXS and SAXS data analysis

For each temperature studied, about 20,000–200,000 single-shot images were collected, filtered (to exclude missed and frozen droplets), and averaged; see Supplementary Note 1. The procedures to obtain the structure factor $S(q)$ from the measured scattering intensity $I(q)$ are detailed in Supplementary Note 1. Note that the liquid structure factors presented herein are normalized to the molecular form factor $f(q)$, and hence represent the center-of-mass arrangement of liquid molecules in reciprocal space. The structure factor first-peak position $q_1(T)$ (in the WAXS patterns) was extracted by fitting a Gaussian function over the range $q \approx 1.7$–$1.9$ Å$^{-1}$. A similar fitting procedure was applied to obtain the $q_1$ structure factor peak positions from the XRD measurements and MD simulations of the glycerol-water solution, as well as from reference structure factor patterns of pure water in literature[7,53]. These data were fitted with Gaussian functions over a slightly larger accessible range $q \approx 1.7$–$2.1$ Å$^{-1}$.

The total SAXS structure factor $S(q)$ is analyzed by following the same procedure employed in ref. 7,50,51. Specifically, $S(q)$ is decomposed as the sum of a normal liquid component, $S_N$, and an anomalous component, $S_A$,

$$S(q) = S_A(q) + S_N(q) \qquad (3)$$

The Percus-Yevick (PY) structure factor for a hard-sphere system, $S_{PY}(q)$, is used to describe the normal component of the structure factor, i.e., $S_N(q) = S_{PY}(q)$. We utilize the PY method provided by the *Jscatter* (1.6.4) package[71,72], $S_{PY}(q, R, \eta)$. The parameters $R$ and $\eta$ are the hard-sphere radius and the volume fraction of the system, respectively, and are obtained by fitting the SAXS curves for $0.15 < q < 0.7$ Å$^{-1}$. To minimize the number of fitting parameters we assume that $R$ is temperature-independent ($R = 1.78 \pm 0.01$ Å); this assumption does not affect the resulting trends.

The excess anomalous scattering $S_A(q)$ is due to critical fluctuations and is described by the Ornstein-Zernike relation[54,57,73]; for a given temperature,

$$S_A(q) = \frac{S_A(0)}{1 + \xi^2 q^2}, \qquad (4)$$

where $\xi$ is the correlation length and $S_A(0)$ is the excess anomalous scattering at $q = 0$. By fitting the $S(q)$ in the SAXS domain using Eq. (3), we additionally obtain the structure factor extrapolated at $q \to 0$, from which the isothermal compressibility of the solution is calculated via Eq. (1).

## X-ray diffraction (XRD) measurements

X-ray diffraction (XRD) measurements of glycerol-water solutions ($\chi_g = 3.2\%$ glycerol mole fraction), made from the same batch solution used for the SACLA XFEL experiments, were conducted using a Bruker D8 VENTURE Single Crystal XRD with a Photon III detector and Mo ($K$-$\alpha$) source. The solution was measured in Kapton capillaries (1 mm diameter) with a 70 mm sample-detector distance. The solutions were studied at temperatures ranging from $T = 295$ K to moderate supercooling $T = 250$ K using a liquid nitrogen-cooled air flow and a 15 min temperature equilibration time. The larger WAXS $q$-range ($q \approx 0.6$-6 Å$^{-1}$) acquired from the XRD measurements was utilized for the accurate normalization of the scattering intensity $I(q)$ to electron units, and conversion to the structure factor $S(q)$ for the SACLA XFEL scattering patterns, as described in Supplementary Note 1.

## Molecular dynamics (MD) simulations

All MD simulations were performed using the GROMACS 2022.5 software package[74]. The CHARMM36 force field[75,76] was used to represent the glycerol molecules and the TIP4P/2005 model was used to model the water molecules[77]. The simulation box contains 320 and 9,680

molecules of glycerol and water, respectively, and the system box dimensions are approximately 69Å × 69Å × 69Å. Periodic boundary conditions apply along the three directions. The simulation time step is $dt = 2$ fs.

MD simulations are performed at $T = 190, 200, 210, \ldots 290$ K. The equilibration of the system is done in two steps: (1) a short 10 ns MD simulation at constant volume (NVT), (ii) followed by another MD simulation performed at constant pressure (NPT) for $0.2$–$1.0\,\mu$s, depending on temperature (from $0.2\,\mu$s at $T = 290$ K to $2.0\,\mu$s at 190K). During equilibration, the temperature is controlled using a Nosé-Hoover thermostat (with a coupling time of 1 ps) while the pressure is kept constant using a Berendsen barostat (with a coupling time of 2 ps). The system is equilibrated at each temperature, independently by monitoring the time dependence of the potential energy (see Supplementary Note 5). At each temperature, equilibration is followed by a 200 ns production run at $P = 1$ bar (see Supplementary Note 5). During the production runs, the temperature and pressure are controlled by a Nosé-Hoover thermostat (with a coupling time of 1 ps) and a Parrinello-Rahman barostat (with a coupling time of 10 ps). This approach allows for efficient pre-equilibration due to the fast convergence of the Berendsen[78] barostat, whereas the Parrinello-Rahman barostat[79,80] combined with the Nosé-Hoover temperature coupling[81], allows to sample accurately density fluctuations as it gives the correct ensemble. The total production run time is longer than the time required for the density autocorrelation function to decay to zero (see Supplementary Fig. 11). Similarly, the total production time is long enough so that both the glycerol and water molecules reach the diffusive regime at which the corresponding mean-square displacement (MSD) increases linearly with time. In addition to checking for any possible sources of uncertainty due to the different starting geometries, we have repeated all simulations three times, starting from independent molecular geometries and repeating the equilibration and production runs, as described above (see Supplementary Note 5). Part of the calculations were performed in a reproducible environment using the Nix package manager together with NixOS-QChem[82].

## Simulated X-ray structure factor

In the MD simulations, the X-ray structure factor $S(q)$ was calculated from the radial distribution function $g(r)$ according to the method described in ref. 22. Specifically,

$$S(q) \simeq 1 + 4\pi\bar{\rho} \int_0^{r_{max}} w(r)r[g(r) - 1]\frac{\sin(qr)}{q}\,dr, \qquad (5)$$

where $\bar{\rho}$ is the number density and $r$ is the radial distance. In Eq. (5) only the heavy atoms are used for the calculation of $g(r)$ (i.e., H atoms are excluded) since these atoms dominate the scattering cross-section at the experimental photon energies considered. The integration interval is limited to $r_{max} = a/2$ where $a$ is the simulation box size. $w(r)$ is a window function introduced to minimize truncation ripples arising from the Fourier transform (Eq. (5)) over a limited integration interval; $w(r)$ is given by[22]

$$w(r) = \begin{cases} 1 - 3\left(\frac{r}{r_{max}}\right)^2, & r < \frac{1}{3}r_{max} \\ \frac{3}{2}\left[1 - \frac{2r}{r_{max}} + \left(\frac{r}{r_{max}}\right)^2\right], & \frac{1}{3}r_{max} < r < r_{max} \\ 0, & r > r_{max} \end{cases} \qquad (6)$$

Eq. (5) is valid in the dilute limit (see details in Supplementary Note 1) and is utilized here as an approximation for the dilute solution (3.2% glycerol mole fraction). For every temperature, the $g(r)$ is computed individually for each frame and then averaged over every 10 frames separated by 100 ps (in total 1 ns). From each averaged $g(r)$, a

corresponding $S(q)$ is determined. The standard error in the $S(q)$ is calculated over the whole production run (200 ns).

## Isothermal compressibility from the MD simulations

The isothermal compressibility $\kappa_T(T)$ is calculated from the MD simulations by evaluating the volume fluctuations ($\delta V$) in the system during the production runs[74,83]. Specifically,

$$\kappa_T = \frac{\langle \delta V^2 \rangle_{\text{NPT}}}{k_B \langle V \rangle \langle T \rangle}, \tag{7}$$

where $\langle V \rangle$ is the time-averaged volume of the system, $k_B$ is the Boltzmann constant, and $\langle T \rangle$ is the time-averaged temperature.

## Calculation of the Local Structure Index (*LSI*)

For a given water molecule $i$, the corresponding local structure index ($LSI_i$) is given by[59–61]

$$LSI_i = \frac{1}{n_i} \sum_{j=1}^{n_i} \left[ \Delta_{ij} - \langle \Delta_{ij} \rangle \right]^2, \tag{8}$$

where $n_i$ is the number of water oxygen neighbors that molecule $i$ has within a O–O cutoff distance $r_c^{OO} = 3.7$ Å. $\Delta_{i,j} = r_{i,j+1} - r_{i,j}$ where $r_{i,j}$ is the distance between the oxygen atoms of molecules $i$ and $j$.

To improve the resolution of the distribution of *LSI* values of the different water molecules, we calculated the *LSI* parameter of the molecules at the so-called inherent structure[61]. For a given instantaneous configuration, sampled during the MD simulation, the inherent structure is the corresponding configuration obtained by minimization of the potential energy of the system. For a given temperature, we obtained 101 configurations evenly sampled from the production run. Potential energy minimization of these configurations was performed using the steepest descent algorithm. The system is considered to reach its inherent structure when the maximum force (gradient in the potential) during the minimization algorithm is smaller than 50 kJ mol$^{-1}$ nm$^{-1}$.

## Data availability

The data are available from the authors upon request. Source data are provided with this paper.

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

## Acknowledgements

F.P. acknowledges financial support by the Swedish National Research Council (Vetenskapsrådet) under Grant No. 2019-05542, 2023-05339 and within the Röntgen-Ångström Cluster Grant No. 2019-06075. F.P. acknowledges the kind financial support from Knut och Alice Wallenberg foundation (WAF, Grant. No. 2023.0052). F.P. acknowledges research is supported by the Center of Molecular Water Science (CMWS) of DESY in an Early Science Project, the MaxWater initiative of the Max-Planck-Gesellschaft, Carl Tryggers (Project No. CTS21:1589) and the Wenner-Gren Foundations (Project No. UPD2021-0144). F.P. acknowledges funding from the European Union's Horizon 2020 research and innovation programme under the Marie Skłodowska-Curie grant agreement No. 101081419 (PRISMAS) and 101149230 (CRYSTAL-X). The experiments were performed at beamline BL3:EH2 of SACLA with the approval of the Japan Synchrotron Radiation Research Institute (Proposal No. 2022B8033). Simulations were performed at the the Sunrise HPC facility supported by the Technical Division at the Department of Physics, Stockholm University (https://doi.org/10.5281/zenodo.TBD) and by resources provided by the Swedish National Infrastructure for Computing (SNIC) at the PDC Center for High Performance Computing, KTH Royal Institute of Technology, partially funded by the Swedish Research Council through grant agreement no. 2018-05973. T.K. acknowledges JSPS KAKENHI (Grant Numbers JP19H05782, JP21H04974, JP21K18944). N.G. is thankful for support from the SCORE Program of the National Institutes of Health under Award No. 1SC3GM139673, the NSF CREST Center for Interface Design and Engineered Assembly of Low Dimensional systems (IDEALS) [NSF Grant Nos. HRD-1547380 and HRD-2112550], and NSF Grant No. CHE-2223461. K.N., M.S., and K.H.K. are supported by the National Research Foundation of Korea (NRF) grant funded by the Korea government (MSIT) (NRF-2020R1A5A1019141).

## Author contributions

S.B. and F.P. conceived, designed and coordinated the experiment with support from T.K. I.A. and F.P. performed the MD simulations and theoretical analysis, with support from N.G. and M.K. S.B., M.F., M.B., A.G., K.N., M.S., K.H.K. and F.P. collected data and participated in the beamtime. S.B., M.F., T.K., F.P., K.N. and M.S. prepared the setup and handled the samples. A.G. and M.B. performed online data processing and analysis, whereas S.B., M.F., K.N., M.S. were responsible for the elog. S.B. and I.A. performed post-beamtime data processing and analysis. The manuscript was written by S.B., I.A., N.G., and F.P. with input from all authors.

## Funding

## Competing interests

The authors declare no competing interests.
