## [Transparent Peer Review file · Nature Communications]

Supercritical density fluctuations and structural heterogeneity in supercooled water-glycerol microdroplets

Corresponding Author: Professor Fivos Perakis

Version 0:

Reviewer comments:

Reviewer #1

(Remarks to the Author)

The paper investigates the structural and thermodynamic properties of supercooled glycerol-water microdroplets using a combination of experimental and computational techniques. The focus is on understanding the impact of glycerol on the behavior of water, particularly its liquid-liquid phase transition (LLPT) and associated density fluctuations.

The study employs experimental techniques such as rapid evaporative cooling, ultrafast X-ray scattering (SAXS and WAXS) at SACLA XFEL, in combination with Molecular dynamics (MD) simulations to complement and validate experimental findings.

The paper provides significant insights into the behavior of supercooled glycerol-water solutions and the influence of glycerol on water's density fluctuations and phase transitions. The combination of experimental and computational techniques is a strong point, but the study would benefit from addressing the following points:

Experimental Design:

1) The use of rapid evaporative cooling and ultrafast X-ray scattering is innovative and allows access to deeply supercooled states. However, the methodology description lacks sufficient details on the reproducibility and control of experimental conditions. For instance, more information on how temperature homogeneity is ensured across microdroplets would be beneficial.

The estimation of droplet temperature based on Knudsen evaporation theory and thermodynamic properties requires rigorous validation. The paper should discuss potential deviations from the theoretical model in the experimental setup, including a discussion on the potential effect of deviations from the estimated temperature.

Computational Parameters:

2) The MD simulations use the CHARMM36 force field for glycerol and TIP4P/2005 model for water. While these are standard choices, the paper should provide a more detailed justification for the selection of these specific models and discuss any known limitations or discrepancies with experimental data.

3) The description of equilibration and production runs in MD simulations is adequate, but the choice of parameters like simulation time, thermostat and barostat settings could be more thoroughly justified. Additionally, potential sources of error or uncertainty in the simulations should be acknowledged.

4) The simulations should be repeated at least three times using different starting geometries to check for reproducibility and determine error bars.

X-ray Structure Factor:

5) The structure factor data obtained from SAXS and WAXS is well-presented. However, the interpretation of the isosbestic

points and their shifts could be expanded. The paper should discuss in more detail how these shifts correlate with the presence of glycerol and what this implies for the local structure of the solution.

6) The comparison between experimental and simulated structure factors is robust, but the observed differences in SAXS intensity should be explored further. Potential reasons for these discrepancies, such as limitations in the MD simulations or experimental uncertainties, should be addressed. Such effects are common for "borderline" systems such as ionic liquids or highly concentrated salt solutions, but in "standard" mixtures of neutral molecular liquids, such a deviation could be a hint to some overly simplified parameter.

Thermodynamic Properties:

7) The analysis of isothermal compressibility (κT) and correlation length (ξ) is insightful. However, the lack of a clear maximum in κT for glycerol-water solutions, contrasted with pure water, raises questions about the completeness of the data. The paper should consider whether additional temperature points around 230 K could provide more conclusive evidence.

8) The power-law fits to κT and ξ are appropriate, but the implications of the fitted parameters (γ and ν) should be discussed in greater depth. Specifically, the paper should elaborate on how these values compare with theoretical predictions and what they reveal about the underlying physical processes.

Perspectives:

9) The study's conclusions about the impact of glycerol on water's density fluctuations and LLPT are well-supported by the data. However, the practical implications for cryopreservation and other applications should be expanded. The paper should discuss how these findings could influence the design of cryoprotectants and what further research is needed to translate these insights into practical solutions.

10) The suggestion for further research into higher concentrations of glycerol and other cryoprotectants is valid. The paper could be improved by providing more specific recommendations for experimental setups or computational studies that could address the current study's limitations. The paper should outline a more detailed plan for how this could be experimentally or computationally investigated, including potential challenges and how they might be overcome.

Given the above, the manuscript could eventually be considered for publication after major revisions from the authors

Reviewer #2

(Remarks to the Author)

Reviewer #3

(Remarks to the Author)

The reviewer thanks the authors for the opportunity to review their paper on Supercritical density fluctuations and structural heterogeneity in supercooled water-glycerol microdroplets. The paper presents extensive data - both experimental and MD on this particular system.

To the reviewer's knowledge, the findings are novel. The paper could benefit from some expansion on why the results are relevant to a broader field - at the moment there is a single sentence mentioning cryopreservation, but this is a distant connection. It would be good if the results were firmly placed in existing knowledge, and how this new knowledge will influence other fields.

Version 1:

Reviewer comments:

Reviewer #1

(Remarks to the Author)

The authors have addressed my suggestion in an effective way.

The manuscript can be accepted as is

Reviewer #2

(Remarks to the Author)

All of my concerns have been positively addressed in the revision.

Rebuttal letter for “Supercritical density fluctuations and structural heterogeneity in supercooled water-glycerol microdroplets” (NCOMMS-24-26629).

We appreciate the time and effort each of the three reviewers dedicated to providing insightful feedback on ways to strengthen our paper and the overall positive evaluation of our work. Here, we provide a point-by-point response to the reviewers’ comments, including a detailed list of changes to the manuscript.

Reviewer #1 (Remarks to the Author):

The paper investigates the structural and thermodynamic properties of supercooled glycerol-water microdroplets using a combination of experimental and computational techniques. The focus is on understanding the impact of glycerol on the behavior of water, particularly its liquid-liquid phase transition (LLPT) and associated density fluctuations. The study employs experimental techniques such as rapid evaporative cooling, ultrafast X-ray scattering (SAXS and WAXS) at SACLA XFEL, in combination with Molecular dynamics (MD) simulations to complement and validate experimental findings. The paper provides significant insights into the behavior of supercooled glycerol-water solutions and the influence of glycerol on water's density fluctuations and phase transitions. The combination of experimental and computational techniques is a strong point, but the study would benefit from addressing the following points:

We thank the reviewer for providing interesting and detailed remarks about our work. We have revised our manuscript to address the reviewer’s comments which helped improve our paper significantly.

Experimental Design:

1) The use of rapid evaporative cooling and ultrafast X-ray scattering is innovative and allows access to deeply supercooled states. However, the methodology description lacks sufficient details on the reproducibility and control of experimental conditions. For instance, more information on how temperature homogeneity is ensured across microdroplets would be beneficial. The estimation of droplet temperature based on Knudsen evaporation theory and thermodynamic properties requires rigorous validation. The paper should discuss potential deviations from the theoretical model in the experimental setup, including a discussion on the potential effect of deviations from the estimated temperature.

We have added the following section in the supplementary information, related to reproducibility and control of experimental conditions:

(page 4) “see Supplementary Information for details on the calculations of $S(q)$, reproducibility, and control of experimental conditions.”

(SI) “1.4 Reproducibility and control of experimental conditions.

The reproducibility of the measurements can be confirmed by the data obtained during two independent runs, illustrated in Fig. S4. During run 1, we were able to reach temperatures from 259.8 K down to 232.1K, whereas in run 2 we could access the range from 259.3 K down to 229.3 K. We observe that the data is highly reproducible within the experimental error bars as can be seen by the comparison in the $S(q)$ line shape (Fig. S4A and B), but also from the temperature dependence of the q_1 peak position and isothermal compressibility κ_T (Fig. S4C and D). As an additional independent experimental control, we have also included data collected with a tabletop X-ray diffractometer (empty circles in Fig. S4C), which aligns well with the XFEL data. This consistency among various independent measurements validates our results and indicates that the observed trends are independent of changes in the experimental configuration.

Fig. S4 | Reproducibility and control of experimental results. (A-B) The structure factor $S(q)$ obtained during two independent measurements (run 1 and run 2). (C-D) The corresponding $S(q)$ first-peak position, q_1 , and isothermal compressibility, κ_T obtained for the two runs. “

In addition, we have added the following section to elaborate on the droplet temperature estimation:

(page 19) “The Knudsen evaporation model for droplet temperature estimation has been examined in detail previously, both experimentally based on XFEL [7, 9] and Raman [73] measurements, as well as from MD simulations [74] (see also Supplementary Section S3). ”

(SI): “The Knudsen evaporation model employed in this work for the estimation of the droplet temperature has been validated in prior studies, including both experimental approaches [6, 9], as well as molecular dynamics (MD) simulations [10]. Given the small deviation observed within independent datasets, we conclude that the temperature of the microdroplet for a given travel time is relatively homogeneous, in agreement with previous studies [6, 9].”

“We estimate that the main source of experimental uncertainty in the droplet temperature estimation by Knudsen theory is likely the droplet size. This uncertainty arises from the size determination from 2D microscope images, where the cross-section of the 3D droplets depends on the camera sharpness and focusing. Based on the microscope images, we estimate that the uncertainty in the droplet diameter is, at most, $\sigma_d = \pm 3 \mu\text{m}$, where $d_0 = 18.7 \mu\text{m}$ is the determined droplet diameter. Such deviations in droplet size would result in different droplet temperatures calculated with Knudsen evaporation theory. The resulting uncertainty in the temperature decreases upon cooling, from approximately $\sim 3 \text{ K}$ at $T_0 = 260 \text{ K}$ to $\sim 1 \text{ K}$ at $T_0 = 230 \text{ K}$ (Fig. S6). Therefore, we conclude that the uncertainty in droplet size and temperature should not significantly alter the observed temperature trends of the experimental q_1 position and compressibility κ_T . Furthermore, the high level of data reproducibility suggests that the

underlying physical processes influencing the droplet behavior remain robust despite any minor variations in droplet temperature.”

Fig. S6 | *Droplet temperature estimation calculated with the Knudsen evaporation theory for droplets with size $d_0=18.7 \mu\text{m}$. The error-bars depict the temperature uncertainty upon cooling, for a droplet size uncertainty of $\sigma_d=\pm 3 \mu\text{m}$.*

Computational Parameters:

2) The MD simulations use the CHARMM36 force field for glycerol and TIP4P/2005 model for water. While these are standard choices, the paper should provide a more detailed justification for the selection of these specific models and discuss any known limitations or discrepancies with experimental data.

We follow the reviewer’s suggestion and add the following discussion in the SI:

(SI) *“The CHARMM36 force field [21, 22] was used to represent the glycerol molecules and the TIP4P/2005 model was used to model the water molecules [23]. The CHARMM force field is a well-known additive, all-atom force field that has been used extensively in the past to study proteins, nucleic acids, lipids, and carbohydrates. The CHARMM force field combined with the TIP4P/2005 water model have been used in the past to study the structural, dynamical, and thermodynamic properties of glycerol-water [24-29]. Our rationale for employing the TIP4P/2005 water model is that it reproduces very well the properties of bulk water and many of the properties of glycerol-water mixtures [26-28]. In particular, the TIP4P/2005 water model reproduces qualitatively well the anomalous properties of water, including the compressibility maximum at 1 bar, and it exhibits a liquid-liquid critical point [30].*

We note that our results from MD simulations, based on the CHARMM36 force field and TIP4P/2005 water model, are in very good agreement with the experiments. To show this we consider a glycerol-water solution with $\chi_g = 3.2\%$ glycerol mole fraction and compare the location of the first peak of the structure factor, q_1 , obtained from experiments and MD simulations. As shown in Fig. S9A, the values of q_1 obtained from experiments and MD simulations are in very good agreement with each other; particularly at $T > 255$ K. At lower temperatures, the values of q_1 obtained from MD simulations appear to deviate slightly from the experimental values. A similar trend is observed in the values of q_1 reported from MD simulations of bulk water [30]. Since the q_1 position correlates with the tetrahedrality fraction of water [6], this observation indicates that TIP4P/2005 can underestimate the tetrahedral local coordination at ambient pressure.

The isothermal compressibility κ_T of the glycerol-water solution ($\chi_g = 3.2\%$) calculated from (i) the experimental structure factor measured in the XFEL experiment and (ii) the MD simulations are included in Fig. S9B. As for the case of pure TIP4P/2005 water, the MD simulations of the

glycerol-water mixture reproduce the qualitative increase of κ_T upon cooling. However, our MD simulations underestimate the value of κ_T relative to the experiments. This is not surprising since most empirical rigid/flexible water models, including the TIP4P/2005 model, underestimate the values of κ_T of bulk water at low temperatures [30]. Despite this limitation inherent to most of the available water models, our results from MD simulations in Fig. S9B clearly show a maximum in the κ_T of the glycerol-water solution ($\chi_g = 3.2\%$) at $T \approx 223$ K, at temperatures slightly below the lowest temperature accessed in our experiments.

Fig. S9 | Comparison of the structure factor first-peak position, $q_1(T)$, obtained from the XFEL and XRD experiments (blue filled and empty circles respectively), as well as from MD simulations (orange triangles) of the studied glycerol-water solution ($\chi_g = 3.2\%$ glycerol mole fraction). The agreement between experiments and MD simulations at $T > 229$ K is remarkably good. (B) Temperature dependence of the isothermal compressibility κ_T of the glycerol-water solution calculated from the experimental structure factor measured in the XFEL experiment (blue circles) and from the MD simulations (orange triangles). As for the case of pure TIP4P/2005 water, the MD simulations of the glycerol-water mixture reproduce the qualitative increase of κ_T upon cooling although they underestimate the value of κ_T relative to the experiments. Note that MD simulations show a clear maximum in the κ_T of the solution at $T \approx 223$ K, slightly below the lowest temperature accessed in our experiments.

3) The description of equilibration and production runs in MD simulations is adequate, but the choice of parameters like simulation time, thermostat and barostat settings could be more thoroughly justified. Additionally, potential sources of error or uncertainty in the simulations should be acknowledged.

We follow the reviewer's suggestion and update the methods section as follows.

(page 21): "MD simulations are performed at $T=190, \sim 200, \sim 210, \dots, 290$ K. The equilibration of the system is done in two steps: (1) a short 10 ns MD simulation at constant volume (NVT), (ii) followed by another MD simulation performed at constant pressure (NPT) for 0.2-2.0 μs , depending on temperature (from 0.2 μs at $T=290$ K to 2.0 μs at $T=190$ K). During equilibration, the temperature is controlled using a Nose-Hoover thermostat (with a coupling time of 1 ps) while the pressure is kept constant using a Berendsen barostat (with a coupling time of 2 ps). The system is equilibrated at each temperature, independently by monitoring the time dependence of the potential energy (see Supplementary Section S5). At each temperature, equilibration is followed by a 200 ns production run at $P=1$ bar (see Supplementary Section S5). During the production runs, the temperature and pressure are controlled by a Nose-Hoover thermostat (with a coupling time of 1 ps) and a Parrinello-Rahman barostat (with a coupling time of 10 ps). This approach allows for efficient pre-equilibration due to fast convergence of the Berendsen barostat [82], whereas the Parrinello-Rahman barostat [83, 84]

combined with the Nosé-Hoover temperature coupling [85], allows to sample accurately density fluctuations as it gives the correct ensemble. The total production run time is longer than the time required for the density autocorrelation function to decay to zero (see Supplementary Fig. S11). Similarly, the total production time is long enough so that both the glycerol and water molecules reach the diffusive regime at which the corresponding mean-square displacement (MSD) increases linearly with time. In addition to checking for any possible sources of uncertainty due to the different starting geometries, we have repeated all simulations three times, starting from independent molecular geometries and repeating the equilibration and production runs, as described above (see Supplementary Section S5).”

and added in the SI:

(SI): “The equilibration time ranged from 200 ns at $T=290$ K to 2 μ s at 190K based on time-dependence of the potential energy shown in Fig. S11. The corresponding simulation time for the production runs was 200 ns and was determined based on the characteristic correlation time obtained from the density-density correlation functions, as shown in Fig. S11. This approach ensures full decorrelation within the production run and allows complete sampling of the density fluctuations at lower temperatures. In addition, we confirm that the mean-square displacement of glycerol and water becomes a linear function of time within the simulation time for the production runs, i.e., both the glycerol and water molecules reach the diffusive regime (see Fig. S11C and Fig. S11D).

Fig. S11 | (A) Potential energy as a function of time during the equilibration of all MD simulations of the glycerol-water solution at $T=190$ K. The horizontal black dashed line represents the average of the last 100 data points and serves as a visual guide. (B) Density-density time correlation of the MD simulations (production runs) of water-glycerol for different temperatures shown in the legend. We

observe that full decorrelation occurs within 200 ns, which is the simulation time of the production runs for each temperature. (C) The mean-square displacement of water (with reference to the oxygen, O_w) and (D) that of glycerol (with reference to the central carbon, C_3).

4) The simulations should be repeated at least three times using different starting geometries to check for reproducibility and determine error bars.

We have repeated the simulations three times using different starting configurations, including the full equilibration and production runs, as suggested by the referee, and added a comparison in the SI:

Fig. S12 | Results from three MD simulations of a glycerol-water solution at $P=1$ bar ($\chi_g = 3.2\%$); MD simulations start from a different starting configuration. (A) Position of the structure factor first peak, $q_1(T)$, and (B) isothermal compressibility, $\kappa_T(T)$, as a function of temperature. The values of $q_1(T)$ and $\kappa_T(T)$ are reproducible among the three runs within the error bars. The MD simulation results reported in the main manuscript are the average values of the three runs shown here.

X-ray Structure Factor:

5) The structure factor data obtained from SAXS and WAXS is well-presented. However, the interpretation of the isosbestic points and their shifts could be expanded. The paper should discuss in more detail how these shifts correlate with the presence of glycerol and what this implies for the local structure of the solution.

Following the reviewer's suggestion, we have expanded the discussion on the isosbestic point:

(page 5): "Interestingly, there is an isosbestic point in the $S(q)$ shown in Fig. 2A located at $q \approx 0.5 \text{ \AA}^{-1}$. While it is not evident whether there is an underlying physical/chemical reason for the existence of this isosbestic point in the $S(q)$, such an isosbestic point defines a T -independent wavevector q that can be used as a useful for future SAXS studies. The isosbestic point in $S(q)$ is a consequence of the increase in $S(0)$ upon cooling, which is due to the increase in thermal compressibility as the temperature of the solution is lowered. Since the value of $S(q)$ is practically T -independent, the increase of $S(0)$ upon cooling leads to an isosbestic point that barely shifts with decreasing temperature. An isosbestic point is also found in the $S(q)$ of pure water, located at $q \approx 0.4 \text{ \AA}^{-1}$ [7]. MD simulations using TIP4P/2005 indicate an isosbestic point at $q \approx 0.25 \text{ \AA}^{-1}$ at $P = 1$ bar, which shifts to $q \approx 0.4 \text{ \AA}^{-1}$ upon increasing pressure at $P = 1$ kbar [49]. Accordingly, adding glycerol shifts the isosbestic point of the $S(q)$ towards higher q -values, at $q \approx 0.5 \text{ \AA}^{-1}$, which is consistent with the overall shift in the $S(q)$ of water, towards lower values of q , with the addition of glycerol. A similar shift of the isosbestic point of the $S(q)$ has been measured in the SAXS of NaCl-water solutions [50, 51], resembling the trend found in computer simulations of pure water with increasing pressure [22]."

6) The comparison between experimental and simulated structure factors is robust, but the observed differences in SAXS intensity should be explored further. Potential reasons for these discrepancies, such as limitations in the MD simulations or experimental uncertainties, should be addressed. Such effects are common for "borderline" systems such as ionic liquids or highly concentrated salt solutions, but in "standard" mixtures of neutral molecular liquids, such a deviation could be a hint to some overly simplified parameter.

The reviewer raises an important point. We follow the reviewer's suggestion and expanded the discussion about the observed differences in SAXS between the experiment and MD simulations.

(page 6): *“On comparing the SAXS structure factors from experiments and MD simulations (see insets, Fig. 2), we note two main differences. Firstly, there is a slight shift in the q -position of the experimental and simulated isosbestic points of $S(q)$ (in the SAXS range). This is in agreement with previous observations in the $S(q)$ of pure water obtained from experiments/MD simulations [22]. Secondly, there is a small difference in the absolute intensity of the SAXS curves at small q . For example the minimum of $S(q)$ in Fig. 2A increases from $S(q) \approx 0.055$ to $S(q) \approx 0.060$ as the temperature decreases from $T \approx 260$ K to $T \approx 230$ K. Instead, in Fig. 2B, the minimum of $S(q)$ increases from $S(q) \approx 0.030$ to $S(q) \approx 0.040$ (for the same T -interval). This discrepancy is likely due to the difference in the $S(0)$ between experiments and MD simulations observed also for pure water [49], related to the fact that TIP4P/2005 underestimates the compressibility, (see Supplementary section S5 for a direct comparison and detailed discussion). A small difference in the vertical offset and scaling of the SAXS curves can also result from the dilute-limit approximation used for the calculation of the $S(q)$ from the MD simulations (see Supplementary Section~S1) or from experimental uncertainties arising from the background subtraction (see Supplementary Section S1.1).”*

Thermodynamic Properties:

7) The analysis of isothermal compressibility (κ_T) and correlation length (ξ) is insightful. However, the lack of a clear maximum in κ_T for glycerol-water solutions, contrasted with pure water, raises questions about the completeness of the data. The paper should consider whether additional temperature points around 230 K could provide more conclusive evidence.

Based on previous studies, the temperature for the $\kappa_T(T)$ maximum in pure water ($T=230$ K), is reproduced well by MD simulations (TIP4P/2005, $T=234$ K). For the glycerol-water solutions, the fact that we do not observe any κ_T maximum at $T=230$ K, would be consistent with the MD simulations which indicate that the κ_T maximum is shifted to $T=223$ K. In order to explore the possible κ_T maximum experimentally one would need to reach temperatures $T < 223$ K, which is highly non-trivial. In the new version of the manuscript. we have addressed this point and added a description of suggested experimental approaches to boost the data quality for reaching lower temperatures with finer temperature spacing (see response to question 10 below)

Despite the difficulty in accessing the κ_T -maximum in the experiments of the water-glycerol solutions, we further analyzed the current dataset. Our analysis, based on the deviation of κ_T from the power-law model, provides indications of the presence of a κ_T maximum in the solution at $T < 230$ K. We have added the following discussion:

(page 12): *“In order to further explore whether the experimental $\kappa_T(T)$ data provide any indications of a maximum in the $\kappa_T(T)$, we analyze the goodness of fit for the power-law model, based on the coefficient of determination R^2 . The power-law model predicts a divergence at $T=T_s$, where the $\kappa_T(T)$ would be infinite. Approaching the Widom line, it is expected that the $\kappa_T(T)$ would deviate from the power-law prediction in the proximity of the $\kappa_T(T)$ maximum [56]. Fig. S8A shows the power-law fit for the experimental $\kappa_T(T)$ data, including the whole temperature range (dashed line) and by excluding the κ_T data point at $T=230$ K (solid line). Based on the R^2 , we observe that the best fit to a*

power-law is obtained by excluding the κ_T data point at $T = 230\text{K}$ (solid line). Hence, the data point for $\kappa_T(T)$ at $T=230\text{ K}$ deviates from the power-law behavior. We validate this approach by performing a similar analysis on the MD simulations, shown in Fig. S8B. Again, we observe a similar behavior, whereby excluding the $\kappa_T(T)$ data at $T \leq 230\text{K}$ (solid line) provides a significantly better agreement ($R^2 = 0.992$) with a power law. The fit to all values of $\kappa_T(T)$ including the $T = 230\text{K}$ data point is shown by the dashed line ($R^2 = 0.936$). This result indicates that the $\kappa_T(T)$ data deviate from the power-law at $T = 230\text{K}$. We note that the data at $T=220\text{K}$ deviate even further from the power-law behavior, as this is the temperature where the $\kappa_T(T)$ maximum is observed.

Fig. S8 | Comparison of the isothermal compressibility $\kappa_T(T)$ of glycerol-water obtained from (A) the experiment and (B) the MD simulations. The lines depict power-law fits for different temperature ranges (dashed line, all temperatures; solid line, $T > 230\text{K}$). Based on the goodness of the fit (R^2 shown in the legend) we observe that both experimental and MD data indicate deviation from the power law behavior at $T \leq 230\text{K}$.

It should be noted here, that the deviation at $T \leq 230\text{K}$ appears larger for the MD simulation than in the experiment, likely due to limitations of the MD model. TIP4P/2005 model underestimates the amplitude of the $\kappa_T(T)$ maximum for pure water. Indeed, a shift in pressure results in better agreement between the results of MD simulations of TIP4P/2005 water and experiments which can be explained by considering that the location of the liquid-liquid critical point in TIP4P/2005 water is shifted, in the P - T plane, relative to the corresponding location of the liquid-liquid critical point of real water. This effect is also seen in the broader $\kappa_T(T)$ maximum of TIP4P/2005 water, compared to experiments [21], which implies that any deviation observed for the power-law behavior would be more significant for the simulation than the experimental data, as observed in Fig. S8B.

8) The power-law fits to κ_T and ξ are appropriate, but the implications of the fitted parameters (γ and ν) should be discussed in greater depth. Specifically, the paper should elaborate on how these values compare with theoretical predictions and what they reveal about the underlying physical processes.

We have expanded the following section following the reviewer's suggestion:

(page 10): "The corresponding exponent and characteristic temperature are $\gamma = 0.36 \pm 0.02$ and $T_{s,\kappa} = 224 \pm 1\text{ K}$. Similar results hold for the correlation length (see Supplementary Fig. S4B), with corresponding exponent and temperatures being $\nu = 0.26 \pm 0.1$ and $T_{s,\xi} = 221 \pm 7\text{ K}$. The power-law fitting parameters are given in Table 1 and are close to the corresponding values reported from experiments in pure water [56].

Sample	γ	ν	ν/γ
Glycerol-water ($\chi_g = 3.2$ mol%)	0.36 ± 0.02	0.26 ± 0.1	0.73 ± 0.4
Pure water [51]	0.40 ± 0.01	0.26 ± 0.3	0.65
Ising model [53]	1.2	0.6	0.5

If the power law behavior in $\kappa_T(T)$ and $\zeta(T)$ was due to an underlying (liquid-liquid) critical point, the power-law exponents, ν and γ , should increase and reach maximum values as the system approaches the critical pressure [58-60]. In the case of the Ising model, the maximum values for the corresponding exponents are $\nu = 0.63$ and $\gamma = 1.24$ with a ratio of $\nu/\gamma = 0.5$ [53–55]. Previous analysis of the power law exponents of those obtained experimentally supercooled water indicate that ratio of the exponents, $\nu/\gamma = 0.65$, which is relatively close to the ratio $\nu/\gamma = 0.51$ that would be expected exactly at a critical point. Analysis of the γ exponents obtained from MD simulations indicate that the various models examined (TIP4P/2005, SPCE, E3B3 and iAMOEBA) underestimate the γ values compared to the experiment [56].

Here we observe that the γ exponent in glycerol-water is lower than expected for the LLCPC, indicating that the system is in the supercritical region and that this apparent critical behavior is associated with approaching the Widom line upon cooling [56]. In addition, the γ value for the glycerol-water solution ($\gamma = 0.36 \pm 0.02$) is lower than pure water ($\gamma = 0.40 \pm 0.01$) indicating possibly that glycerol partially suppresses the critical fluctuations, which is consistent with the relative reduction in the compressibility.”

Perspectives:

9) The study's conclusions about the impact of glycerol on water's density fluctuations and LLPT are well-supported by the data. However, the practical implications for cryopreservation and other applications should be expanded. The paper should discuss how these findings could influence the design of cryoprotectants and what further research is needed to translate these insights into practical solutions.

We followed the reviewer's suggestion and updated the last section in the manuscript:

(page 17): “In conclusion, our results shed light on the influence of glycerol on the local structure of supercooled water and provide evidence that the two-liquid framework of water can be extended to describe the thermodynamic properties of solutions. Alternatively, our findings show how the properties of pure water, and its underlying LLCPC, may affect the properties of aqueous solutions. We find that, even at dilute conditions, the presence of glycerol can partially suppress the collective density fluctuations of pure water and shift its LLCPC towards lower temperatures. The suppression of density fluctuations observed here can be linked to glycerol's cryoprotectant ability to frustrate the low-density fluctuations associated with the formation of the critical nucleus preceding crystallization of the system [70]. The knowledge of the changes in the local water structure and nanoscale fluctuations, with increasing glycerol content, as provided in the present study, are then potentially crucial to design the appropriate cryoprotectant mixtures.

Our results may therefore be important to advance cryopreservation techniques and the design of cryoprotectants that better prevent ice nucleation, a key challenge in cryopreservation. By tuning glycerol concentrations, or combining it with other cryoprotectants like DMSO or ethylene glycol, formulations can be optimized to maximize the suppression of crystallization while maintaining low toxicity [27]. This is particularly relevant for biological applications where ice formation can damage cells and can be beneficial for cryopreserving complex tissues or organs, where ice formation must be avoided [26]. Additionally, understanding the impact of glycerol on water local structure at elevated pressure would benefit the development of high-pressure cryopreservation techniques in combination with glycerol-based solutions that could further enhance control over freezing processes [71].

10) The suggestion for further research into higher concentrations of glycerol and other cryoprotectants is valid. The paper could be improved by providing more specific recommendations for experimental setups or computational studies that could address the current study's limitations. The paper should outline a more detailed plan for how this could be experimentally or computationally investigated, including potential challenges and how they might be overcome.

(page 16): *“Extending our experiments to lower temperatures, $T < 230$ K, to directly observe the maximum in $\kappa_T(T)$ is highly non-trivial. One possible avenue to extend the experimentally accessible temperatures to significantly lower values would be to utilize the high repetition rates of superconducting XFELs (like the European XFEL or LCLS-II), which allow the acquisition of data at kHz to MHz rates (vs 50Hz at SACLA). This approach would enable orders of magnitude more statistics even at very low hit-rates, which can be especially useful for accessing the regime $T < 230$ K, where the hit-rate is on the order of 0.1%. This comes with challenges related to operating the liquid jet at kHz to MHz rates, as well as big data volumes, which have been solved already for serial crystallography [68]. An alternative approach to explore the existence of an LLCPP at elevated pressure would be to prepare high-density amorphous ice samples with different glycerol-water concentrations and perform a temperature-jump experiment (as in Refs. [6, 10, 12]). Possible challenges here relate to establishing accurately the trajectory across the pressure-temperature phase space and creating thin ice samples to ensure homogeneous heating.”*

Given the above, the manuscript could eventually be considered for publication after major revisions from the authors

We thank the reviewer again for the constructive criticism and we hope that we have sufficiently addressed their comments.

Reviewer #2 (Remarks to the Author):

In the context of the liquid-liquid phase transition, this work addresses how the properties of a specific aqueous solution deviate from those of pure water. The aqueous solution is made of glycerol-water at dilute conditions, $\chi_g = 3.2\%$ glycerol mole fraction. The addition of glycerol does not seem to provoke a strong effect on the local structure of the water: both the temperatures of i) equal fraction between low- and high-density liquids and ii) maximum in $d\rho/dT$ are similar to those of pure water. However, while retaining, in principle, the local structure of pure water, experiments show that the aqueous solution presents a lower isothermal compressibility at a given temperature. Moreover, the associated density fluctuations decrease accordingly. Experimentally, a maximum is not observed as it probably requires deeper supercooling. Nevertheless, computer simulations, which agree quite well with the experiments in the regime where they are both applied, present such a maximum at a lower temperature. From computer simulations, the isothermal compressibility of the solution becomes larger than that of pure water at the lowest temperatures.

The noteworthy results are that by adding glycerol in water at $\chi_g = 3.2\%$ glycerol mole fraction, the aqueous solution potentially presents a liquid-liquid phase transition retained from the pure water scenario but occurring at a different region of the P-T phase diagram. Deeper supercooling is required to achieve the same magnitude of density fluctuations. In fact, computer simulations suggest that the maximum of the isothermal compressibility is lower than in pure water.

This work combines different experimental techniques to study microdroplets of a glycerol-water solution at $\chi_g = 3.2\%$ glycerol mole fraction down to 229.3 K. Molecular dynamics simulations with realistic empirical force fields provide support for the measurements. This work can be valuable for aqueous solutions and related fields, particularly in phase equilibria and nucleation studies. The originality of this work is mainly in the deep supercooling achieved for a particular concentration of glycerol in water. Previous literature is discussed.

We thank the reviewer for the positive and detailed feedback of our work. We are also thankful for supporting the publication of our manuscript.

1) Further articles that may be considered are:

1. Murata, K. I., & Tanaka, H. (2013). General nature of liquid-liquid transition in aqueous organic solutions. *Nature communications*, 4(1), 2844.
2. Woutersen, S., Ensing, B., Hilbers, M., Zhao, Z., & Angell, C. A. (2018). A liquid-liquid transition in supercooled aqueous solution related to the HDA-LDA transition. *Science*, 359(6380), 1127-1131.
3. Bachler, J., Fidler, L. R., & Loerting, T. (2020). Absence of the liquid-liquid phase transition in aqueous ionic liquids. *Physical Review E*, 102(6), 060601.
4. Mallamace, F., Corsaro, C., Mallamace, D., Vasi, S., Vasi, C., & Stanley, H. E. (2016). Some considerations on the transport properties of water-glycerol suspensions. *The Journal of Chemical Physics*, 144(1).
5. Espinosa, J. R., Abascal, J. L. F., Sedano, L. F., Sanz, E., & Vega, C. (2023). On the possible locus of the liquid-liquid critical point in real water from studies of supercooled water using the TIP4P/Ice model. *The Journal of Chemical Physics*, 158(20).

We are thankful to the reviewer for bringing these important studies to our attention. We included these references in the new version of the manuscript

2) The conclusions and claims are sufficiently supported. Limitations are acknowledged. I see no flaws in the data analysis, interpretation, and conclusions. However, the computational results are compared with experiments at a certain temperature. Given that the melting temperature of the employed force field is ~ 250 K, maybe it would be more meaningful to compare against supercooling degrees. Even though water contains glycerol at a diluted concentration, the properties of water should still dominate. Indeed, this work explores the hypothesis that the liquid-liquid behavior of water remains in solution at low concentrations of glycerol. For instance, in Fig. 2., the authors note different scales in the y-axis of the insets, where the same temperatures are studied in experiments and

simulations. If one would compare supercooling degrees, then the scales would not be so different. Note that this could worsen the agreement in other instances. However, I generally find it more meaningful to use \approx than actual temperatures in presenting computational results.

We thank the reviewer for raising this interesting point. Motivated by the reviewer's comment, we test below whether the comparison of the $\kappa_T(T)$ obtained from experiments and MD simulations improves by considering the supercooling temperature " $\Delta T = T_m - T$ " instead of the absolute temperature " T " [here, T_m is the melting temperature of TIP4P/2005 water (MD simulations) and real water (experiments)]. As shown in the Fig. S10 below, a comparison of the $S(q)$ as a function of " ΔT " does not significantly improve the agreement between MD simulations and experiments.

We note that while the melting temperature of TIP4P/2005 water is lower than the corresponding experimental value, the thermodynamic properties of liquid and supercooled water are in very good agreement (with no shift in temperature). Indeed, the density of liquid water at 1 bar practically overlaps with the experimental values of equilibrium and supercooled liquid water. We believe that a better comparison between TIP4P/2005 and real water can be achieved by increasing shifting the pressure. For example, as shown in Ref. [49], the values of $\kappa_T(T)$ of TIP4P/2005 water at $P=1$ kbar show good agreement with the experimental values at 1 bar.

We have included the following section based on the referee's suggestion:

(page 6): "Taking into account the difference in melting temperature of real water and TIP4P/2005 ($T_m \approx 250$ K), and thus comparing supercooled degrees instead of the absolute temperature is not sufficient by itself to account for the observed discrepancy (see Supplementary Fig. S10). It has been shown that comparing the experimental data of pure water with MD simulations at elevated pressures yields a more accurate comparison of the SAXS regime and the corresponding compressibility [49]. This effect can be attributed to the fact that the location of the LLC of the TIP4P/2005 model (see Ref. [15, 21-25, 52]) is shifted in pressure-temperature with respect to the LLC estimation in real water [18]."

Fig. S10 | Direct comparison of the structure factor $S(q)$ obtained from experiment (solid line) and MD simulations (dashed line) for (a) similar temperatures near $T = 250$ K and (b) similar supercooling degrees taking into account the melting point for TIP4P/2005 ($T_m \approx 250$ K at $P = 1$ bar).

3) The methodology is sound, meeting the expected standards. I only have one small concern regarding the length of the computer simulations. How do we know that 1,000 ns of equilibration and 200 ns of production are enough? In the suggested article number 5., a similar force field was

employed and the duration of the lowest temperature spanned over 8,000 ns including equilibration and production. Is there a way to be sure that the system has reached equilibrium? In principle, enough details are provided in the methods for the work to be reproduced.

Please see our response to reviewer 1, points 3 and 4. We have extended the simulations to 2 μ s and added a detailed description of how the equilibration is checked, based on the time-dependence of the potential energy. In addition, we explain that the 200 ns simulation time for the production runs is determined based on the density-density correlation function and mean-squared displacement (see response to referee1, point 3). In addition, we observe that all three independent runs (starting from randomized initial configurations and followed by independent equilibration and production runs) give the same structure factor peak position, q_1 , and compressibility values $\kappa_T(T)$ at the lowest temperatures, suggesting that any out of equilibrium effects may be minor (see response to referee 1, point 4).

4) Page 11 line 15: I would say "intermolecular vibrational effects" because TIP4P/2005 is rigid.

We follow the referee's suggestion and modify the sentence in page 11, line 15, accordingly.

5) The kind of liquid that is more prone to surround glycerol is information accessible from computer simulations. Certainly, the relative concentration is expected to play the main role, but is it the only one? For instance, this could be checked when both liquids represent equally 50% of the water.

Following the referee's suggestion, we have updated the following section:

(page 15) "At $T=232$ K, on the other hand, there are nearly equal fractions of HDL-like and LDL-like waters that form highly interpenetrating networks. The rather large percolation of the LDL and HDL domains throughout the system could reflect maximal fluctuations in the proximity to the Widom line. This observation closely coincides with previous simulations of pure water with the TIP4P/2005 model, where a 1:1 distribution between HDL- and LDL-like molecular species was observed at $T\approx 233$ K [65]. At $T=230$ K, we observe that the LSI does not differ whether one examines the bulk-water or that in the proximity of the glycerol (see Supplementary Fig. S13). It should be noted here, as shown in Ref. [67], that these results can depend on the local order parameter used to examine the nature of water in the hydration layer. Our study based on the LSI order parameter suggests that at $T = 230$ K, the local glycerol environment consists 1:1 of HDL/LDL water molecules, which reversely indicates that the system is isocompositional, i.e. LDL and HDL have the same concentration of glycerol, as suggested previously [28]."

Fig. S13 | (A) Site-site partial radial distribution function (RDF) between the central carbon of glycerol (C_3) and the oxygen atoms of water (O_w). The regions corresponding to the first and second hydration shell ($3 \text{ \AA} < r$

< 6.5 Å) and the bulk region ($r > 10$ Å) are shaded in red and blue, respectively. A representation of a glycerol molecule is included in the inset, with the central carbon highlighted. (B) Comparison of the inherent Local Structure Index (LSI) for water in the first/second hydration shell (solid red line) and in the bulk region (dashed blue line).

6) page 15 lines 4-5: a reference is missing.

Thanks, we have updated this section (see response to referee 1, question 9).

Reviewer #3 (Remarks to the Author):

The reviewer thanks the authors for the opportunity to review their paper on Supercritical density fluctuations and structural heterogeneity in supercooled water-glycerol microdroplets. The paper presents extensive data - both experimental and MD on this particular system.

To the reviewer's knowledge, the findings are novel. The paper could benefit from some expansion on why the results are relevant to a broader field - at the moment there is a single sentence mentioning cryopreservation, but this is a distant connection. It would be good if the results were firmly placed in existing knowledge, and how this new knowledge will influence other fields.

We thank the reviewer for the positive feedback and for the comment regarding the relevance of our work in a broader context, including cryopreservation. We have addressed this point above in our response to reviewer #1, point 9.

**REVIEW OF
“SUPERCRITICAL DENSITY FLUCTUATIONS AND STRUCTURAL HETEROGENEITY IN
SUPERCOOLED WATER-GLYCEROL MICRODROPLETS”**

In the context of the liquid-liquid phase transition, this work addresses how the properties of a specific aqueous solution deviate from those of pure water. The aqueous solution is made of glycerol-water at dilute conditions, $\chi_g = 3.2\%$ glycerol mole fraction. The addition of glycerol does not seem to provoke a strong effect on the local structure of the water: both the temperatures of i) equal fraction between low- and high-density liquids and ii) maximum in dq_1/dT are similar to those of pure water. However, while retaining, in principle, the local structure of pure water, experiments show that the aqueous solution presents a lower isothermal compressibility at a given temperature. Moreover, the associated density fluctuations decrease accordingly. Experimentally, a maximum is not observed as it probably requires deeper supercooling. Nevertheless, computer simulations, which agree quite well with the experiments in the regime where they are both applied, present such a maximum at a lower temperature. From computer simulations, the isothermal compressibility of the solution becomes larger than that of pure water at the lowest temperatures.

The noteworthy results are that by adding glycerol in water at $\chi_g = 3.2\%$ glycerol mole fraction, the aqueous solution potentially presents a liquid-liquid phase transition retained from the pure water scenario but occurring at a different region of the P-T phase diagram. Deeper supercooling is required to achieve the same magnitude of density fluctuations. In fact, computer simulations suggest that the maximum of the isothermal compressibility is lower than in pure water.

This work combines different experimental techniques to study microdroplets of a glycerol-water solution at $\chi_g = 3.2\%$ glycerol mole fraction down to 229.3 K. Molecular dynamics simulations with realistic empirical force fields provide support for the measurements. This work can be valuable for aqueous solutions and related fields, particularly in phase equilibria and nucleation studies.

The originality of this work is mainly in the deep supercooling achieved for a particular concentration of glycerol in water. Previous literature is discussed. Further articles that may be considered are:

1. Murata, K. I., & Tanaka, H. (2013). General nature of liquid-liquid transition in aqueous organic solutions. *Nature communications*, 4(1), 2844.
2. Woutersen, S., Ensing, B., Hilbers, M., Zhao, Z., & Angell, C. A. (2018). A liquid-liquid transition in supercooled aqueous solution related to the HDA-LDA transition. *Science*, 359(6380), 1127-1131.
3. Bachler, J., Fidler, L. R., & Loerting, T. (2020). Absence of the liquid-liquid phase transition in aqueous ionic liquids. *Physical Review E*, 102(6), 060601.
4. Mallamace, F., Corsaro, C., Mallamace, D., Vasi, S., Vasi, C., & Stanley, H. E. (2016). Some considerations on the transport properties of water-glycerol suspensions. *The Journal of Chemical Physics*, 144(1).
5. Espinosa, J. R., Abascal, J. L. F., Sedano, L. F., Sanz, E., & Vega, C. (2023). On the possible locus of the liquid-liquid critical point in real water from studies of supercooled water using the TIP4P/Ice model. *The Journal of Chemical Physics*, 158(20).

The conclusions and claims are sufficiently supported. Limitations are acknowledged.

I see no flaws in the data analysis, interpretation, and conclusions. However, the computational results are compared with experiments at a certain temperature. Given that the melting temperature of the employed force field is ~ 250 K, maybe it would be more meaningful to compare against supercooling degrees. Even though water contains glycerol at a diluted concentration, the properties of water should still dominate. Indeed, this work explores the hypothesis that the liquid-liquid behavior of water remains in solution at low concentrations of glycerol. For instance, in Fig. 2., the authors note different scales in the y-axis of the insets, where the same temperatures are studied in experiments and simulations. If one would compare supercooling degrees, then the scales would not be so different. Note that this could worsen the agreement in other instances. However, I generally find it more meaningful to use supercooling degrees than actual temperatures in presenting computational results.

The methodology is sound meeting the expected standards. I only have one small concern regarding the length of the computer simulations. How do we know that 1,000 ns of equilibration and 200 ns of production are enough? In the suggested article number 5., a similar force field was employed and the duration of the lowest temperature spanned over 8,000 ns including equilibration and production. Is there a way to be sure that the system has reached equilibrium?

In principle, enough details are provided in the methods for the work to be reproduced.

Other comments

Page 11 line 15: I would say "intermolecular vibrational effects" because TIP4P/2005 is rigid.

The kind of liquid that is more prone to surround glycerol is information accessible from computer simulations. Certainly, the relative concentration is expected to play the main role, but is it the only one? For instance, this could be checked when both liquids represent equally 50% of the water.

page 15 lines 4-5: a reference is missing.